# Metagenomics of Parkinson's disease implicates the gut microbiome in multiple disease mechanisms

Zachary D. Wallen [1,2], Ayse Demirkan[3], Guy Twa[1], Gwendolyn Cohen [1,2], Marissa N. Dean[1], David G. Standaert [1], Timothy R. Sampson[2,4] & Haydeh Payami [1,2] ✉

Parkinson's disease (PD) may start in the gut and spread to the brain. To investigate the role of gut microbiome, we conducted a large-scale study, at high taxonomic resolution, using uniform standardized methods from start to end. We enrolled 490 PD and 234 control individuals, conducted deep shotgun sequencing of fecal DNA, followed by metagenome-wide association studies requiring significance by two methods (ANCOM-BC and MaAsLin2) to declare disease association, network analysis to identify polymicrobial clusters, and functional profiling. Here we show that over 30% of species, genes and pathways tested have altered abundances in PD, depicting a widespread dysbiosis. PD-associated species form polymicrobial clusters that grow or shrink together, and some compete. PD microbiome is disease permissive, evidenced by overabundance of pathogens and immunogenic components, dysregulated neuroactive signaling, preponderance of molecules that induce alpha-synuclein pathology, and over-production of toxicants; with the reduction in anti-inflammatory and neuroprotective factors limiting the capacity to recover. We validate, in human PD, findings that were observed in experimental models; reconcile and resolve human PD microbiome literature; and provide a broad foundation with a wealth of concrete testable hypotheses to discern the role of the gut microbiome in PD.

Microbiota are necessary for human health. The gut microbiome aid with dietary metabolism, produce essential metabolites such as vitamins, maintain the integrity of the intestinal barrier, inhibit pathogens, and metabolize drugs and toxicants[1]. The gut microbiome regulates development and continued education of the host's immune response and nervous system by producing specific metabolites with far reaching effects such as the maturation and maintenance of microglia in the brain[2]. An imbalance (dysbiosis) in microbiome composition and function can render host prone to disease. Studies in humans and animal models have revealed disease-related dysbiosis in a range of common metabolic (e.g., diabetes), inflammatory (inflammatory bowel disease), neurologic (Parkinson's disease) and developmental disorders (autism)[1].

Parkinson's disease (PD) is a progressively debilitating disorder that affected 4 million Individuals in the year 2005 and is projected to double to 8.7 million individuals by the year 2030[3]. Although historically defined as a movement disorder, PD is a multi-systemic disease[4]. The earliest sign is often constipation which can precede motor signs by decades[4]. Moreover, PD is etiologically heterogenous. Despite the discovery of several causative genes[5], 90 susceptibility loci spanning

[1]Department of Neurology, University of Alabama at Birmingham, Birmingham, AL 35233, USA. [2]Aligning Science Across Parkinson's (ASAP) Collaborative Research Network, Chevy Chase, MD 20815, USA. [3]Surrey Institute for People-Centred AI, University of Surrey, Guildford, Surrey GU2 7XH, UK. [4]Department of Cell Biology, Emory University School of Medicine, Atlanta, GA 30329, USA. ✉e-mail: haydehpayami@uabmc.edu

the human genome[6], and multiple environmental risk factors[7], the vast majority of PD remains idiopathic. It is speculated that PD is caused by various combinations of genetic susceptibility and environmental triggers, although no causative combination has yet been identified.

Braak's hypothesis[8] that non-familial forms of PD start in the gut by a pathogen is gaining increasing support. The connection between PD and the gastrointestinal (GI) system, including constipation, compromised gut barrier, and inflammation, has long been established. Alpha-synuclein pathology has been detected in the gut of persons with PD at early stages[9], and there is evidence from imaging studies that in some cases pathology may start in the gut and spread to the brain[10]. In mice, it was shown that alpha-synuclein fibrils injected into gut induce alpha-synuclein pathology which spreads from gut to brain, and that vagotomy stops the spread[11]. In parallel, large epidemiological studies have shown that persons who had complete truncal vagotomy decades earlier had substantially reduced incidence of PD later in life[9].

With accumulating evidence implicating the gut as an origin of PD, and the newly gained appreciation for the involvement of gut microbiome in chronic diseases, there has been increasing interest in decoding the connection between the gut microbiome and PD. In mice overexpressing the human alpha-synuclein gene, we have shown that gut microbiome regulates alpha-synuclein-mediated pathophysiologies[12]. In another genetic model of PD (Pink1[-/-]), intestinal infection with Gram-negative bacterial pathogens was shown to elicit an immune reaction that leads to neuronal degeneration and motor deficits, and which can be reversed with the PD-medication, L-dopa[13]. In addition, we and others have found that, curli, an amyloidogenic protein produced by Gram-negative *Escherichia coli*, induces alpha-synuclein aggregation and accelerates disease in the gut and neurodegeneration in the brain[14–16]. We have detected overabundance of opportunistic pathogens in the gut microbiomes of individuals with PD[17]. Collectively, experimental and human studies support Braak's hypothesis that intestinal infection may act as a triggering event in PD, but it is yet to be proven that pathogens in human gut cause PD. Studies conducted on human fecal samples have all found evidence of dysbiosis in PD gut microbiome but results on specific microorganisms that drive the dysbiosis have been mixed[18–20]. Human studies of PD and microbiome have had limited sample sizes, and all except two[21,22] were based on 16S rRNA gene amplicon sequencing (henceforth, 16S) which limits resolution to genus-level. Metagenomics (study of all genetic material sampled from a community) is an emerging field in medical science. With deep shotgun metagenomic sequencing, the microbiome can now be studied in large-scale human studies at high resolution of species and genes.

Here we present a large-scale metagenomics analysis of PD gut microbiome. This study was designed and executed by a single team of investigators (NeuroGenetics Research Consortium, NGRC), enabling complete control to employ state-of-the-art methods and ensure uniformity from start to end. We confirm the common findings from prior studies, resolve them to species level and solve the inconsistencies in the literature. In addition, owing to large sample size and deep shotgun sequencing, we generate a vast amount of new information. We show wide-spread dysbiosis in PD microbiome, identify species that drive the dysbiosis, and by functional profiling, nominate microbial genes and pathways in the gut that may contribute to PD mechanisms.

## Results and discussion

### A large, newly enrolled, and uniformly assessed cohort

As outlined in STORMS[23] flowchart (Fig. 1), the study included a newly enrolled cohort of 490 persons with PD and 234 neurologically healthy controls (NHC). The sample size is comparable to Human Microbiome Project (HMP) which included 100 individuals with inflammatory bowel disease, 106 individuals with pre-diabetes and 242 healthy adults ages 18-40 years old[24]. Defined by self-reported biological sex,

52% of subjects were men, 48% were women. 97% of PD cases and 93% of NHC were over 50 years old. The older ages of the controls in this study (mean $65.8 \pm 8.8$) and their neurologically healthy status is a unique addition to the publicly available datasets. All subjects were from a single geographic area in the Deep-South United States (US), minimizing confounding by geographic variation. Fifty-five percent of NHC were spouses and shared environment with PD. Using uniform methods, we collected extensive metadata (Supplementary Fig. 1) and a stool sample from each subject, extracted DNA from stool and conducted deep shotgun sequencing achieving average 50 M raw reads/sample. This dataset is new and is publicly available.

### Subject characteristics and metadata

Data on 53 variables were analyzed to characterize the subjects, and to identify disease-associated variables that could potentially confound downstream metagenomics analyses (Table 1). GI problems, which are well-known features of PD, were readily evident in this cohort. Constipation was more prevalent in PD cases (odds ratio (OR) = 6.1, 95% confidence interval (CI) = 3.9–10, $P = 2E–19$ for chronic constipation, $P = 3E–6$ for Bristol Chart score), and PD cases reported more GI discomfort than NHC (OR = 2.8, 95% CI = 1.8–4.2, $P = 3E–7$). Compared to NHC, PD cases had diminished intake of alcohol (OR = 0.6, 95% CI = 0.4–0.8, P = 3E–4) and foods in all five categories (fruits/vegetables, animal products, nuts, yogurt, and grains), all reaching significance (OR = 0.6–0.7, P = 0.002–0.05) except grains (OR = 0.8, 95% CI = 0.6–1.1, $P = 0.2$). Use of laxatives (OR = 3.8, 95% CI = 2.4–6.4, $P = 7E–10$), pain medication (OR = 1.6, 95% CI = 1–2.5, P = 0.04), sleep aid (OR = 2, 95% CI = 1.4–2.9, P = 7E–5), and medication for depression/anxiety/mood (OR = 2.1, 95% CI = 1.4–3, P = 7E–5) were more common in PD than NHC. Probiotic supplement use was more common in NHC than PD (OR = 0.6, 95% CI = 0.4–0.9, P = 0.02), which is noteworthy because as the data will show, *Bifidobacterium* and *Lactobacillus* species, which are common constituents of commercial probiotics, were more abundant in PD than NHC metagenomes. Variables that differed in PD vs. NHC were evaluated as potential confounders in downstream metagenomics analyses.

### PD and NHC metagenomes differ from global scale to species-resolution

Inter-individual difference in the global composition of the gut metagenome, beta-diversity, was significantly different in PD vs. NHC ($P < 1E–4$, tested using permutational multivariate analysis of variance (PERMANOVA)). Test of dispersion was also significant ($P < 1E–4$, tested using permutational analysis of multivariate dispersion (PERMDISP)) indicating that the difference in global composition of PD vs. NHC is driven primarily by differences in dispersion, rather than differences in spatial medians. Principal component analysis (PCA) plots also show that PD metagenomes were visibly more dispersed than NHC (Supplementary Fig. 2 a, b). Results were robust when rare taxa (present in <5% of samples) were excluded, yielding $P < 1E–4$ for both PERMANOVA and PERMDISP. Greater dispersion across PD metagenomes may reflect the heterogeneity of PD. PD is not one disease, and each disease mechanism may have a different microbiome signature.

At enterotype level, 284 PD and 166 NHC were confidently classified as one of three enterotypes: *Prevotella*, *Firmicutes* and *Bacteroides* (Supplementary Fig. 2c). The overall enterotype distribution in PD was different from NHC ($P = 1E–4$). Sequential testing identified *Firmicutes* enrichment as the primary driving force ($X^2(2)=44.4$), and depletion of *Prevotella* as secondary as it remained significant ($X^2(1)=3.7$) after removing *Firmicutes*. Effect sizes were OR = 2.4, 95% CI = 1.5–3.9, $P = 1E–4$, for *Firmicutes*, and a non-significant OR = 0.66 (95% CI = 0.3–1.2, P = 0.1) for *Prevotella* after adjusting for *Firmicutes*' effect.

Next, we attempted to identify the disease-associated species. Including Bacteria, Archaea and Eukarya, we identified a total of 2270 species that mapped to at least one marker gene, 719 of which

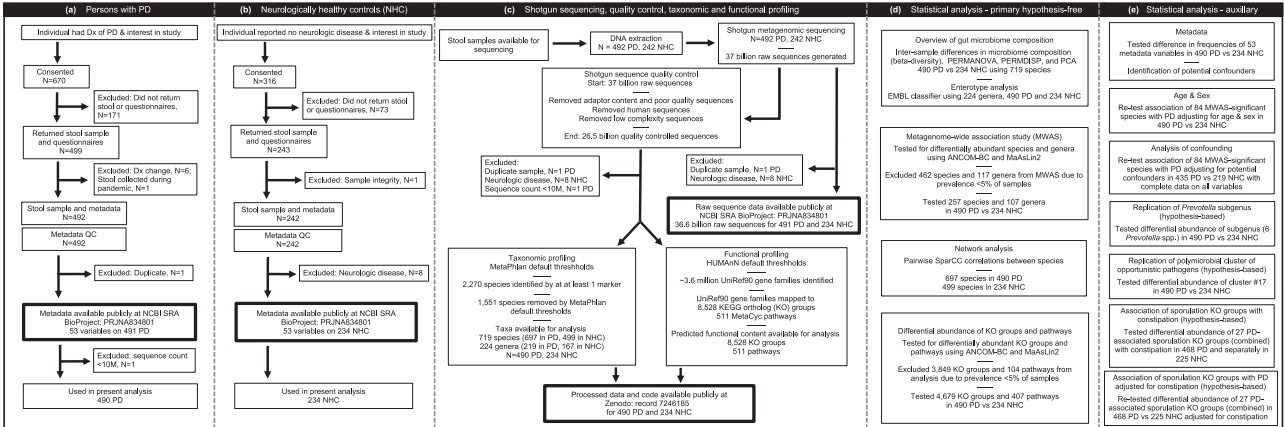

**Fig. 1 | The STORMS flowchart.** Following reporting guidelines for human microbiome research[23], we show the step-by-step process by which the study was conducted, starting with subject selection, enrollment and data collection for persons with PD (**a**) and neurologically healthy controls (**b**), sequencing and bioinformatics pipeline (**c**), and statistical analyses (**d, e**).

passed stringent bioinformatic quality control (QC) thresholds of MetaPhlAn3 (default parameters: t = rel_ab, perc_nonzero=0.33, statq=0.2, avoid_disqm=FALSE, stat=tavg_g, min_cu_len=2000, unknown_estimation=FALSE). 257 species were present in ≥5% of subjects, i.e., 35% which is closely comparable to another large study[25]. We conducted an unbiased metagenome-wide association study (MWAS) to test differential abundances of species in PD vs. NHC, using two statistical methods (MaAsLin2 and ANCOM-BC). We also conducted MWAS at genus level to be able to interpret our results in the context of the existing literature. The full MWAS results are provided in Supplementary Data 1 and 2. We nominated a species or genus as PD-associated if it achieved significance by MaAsLin2 and ANCOM-BC (i.e., false discovery rate (FDR) < 0.05 by one and FDR≤0.1 by the other). By this stringent definition, 84 of 257 species tested were associated with PD: 64 achieved FDR < 0.05 by both MaAsLin2 and ANCOM-BC, 10 had FDR < 0.05 by MaAsLin2 and FDR≤0.1 by ANCOM-BC, and 6 had FDR < 0.05 by ANCOM-BC and FDR≤0.1 by MaAsLin2 (Fig. 2). Of the 84 PD-associated species 55 were enriched and 29 were depleted in PD (Table 2, Fig. 3, Supplementary Fig. 3). Of the 107 genera tested, 34 were associated with PD: 23 were enriched and 11 were depleted in PD (Table 3). Thus, whether measured at species or genus level, the dysbiosis in the PD gut microbiome appears to involve about 30% of tested taxa.

Effect sizes were large (Fig. 3). At one end of the spectrum, *Bifidobacterium dentium* was elevated by 7-fold, *Actinomyces oris* by 6.5-fold, *Streptococcus mutans* by 6-fold. At the other end of the spectrum, *Roseburia intestinalis* was reduced by 7.5-fold, and *Blautia wexlerae* by 5-fold. Overall, 36% (30 of 84) of PD-associated species had higher than two-fold change in abundance by both MaAsLin2 and ANCOM-BC, reflecting a 100% to 750% increase or decrease in PD vs. NHC.

The primary aim of this study was to generate a full, unaltered view of the dysbiosis in PD gut microbiome. To that end, MWAS and downstream analyses were adjusted only for technical variables (i.e., total sequence count per sample, and stool collection kit). Adjusting for variables like constipation, which are intrinsic to PD, would mask their effect in the full view that we are aiming for. However, as secondary analyses, we investigated effects of age and sex, and explored confounding by extrinsic variables that had differing frequencies/distributions in PD vs. NHC, namely intake of alcohol, laxative, probiotics, antihistamines, depression/anxiety/mood medication, pain medication, and sleep aid (Table 1). We tested association of the 84 species, that had emerged from MWAS, with PD while including age and sex in the model, and all species retained significance for association with PD at FDR < 0.1 (Supplementary Data 3). Next, we re-tested the 84 species in a model that included the extrinsic variables (Supplementary Data 4). Statistical

power was substantially reduced when testing all variables simultaneously, to a degree that species that were rare and had smaller analytic sample sizes lost signal all together. Nonetheless, we confirmed the association of PD with 62 of 84 species (Table 2): 32 species were associated only with PD, while the other 30 were associated with PD and one or more other variables, most commonly alcohol (avoided by PD) or laxatives (frequently used by PD).

## Network analyses reveal clusters of co-occurring and competing species

We calculated pairwise correlation (r) in species abundances for PD metagenome using all 697 species detected in PD (Supplementary Data 5), and for NHC metagenome using all 499 species detected in NHC (Supplementary Data 6). We observed both positive and negative correlations throughout the metagenome, including PD-associated species. Positive correlation among species suggests they tend to co-occur in the same sample and their abundances rise or fall together, while negative correlation suggests that, within a sample, an increase in the abundance of one species correlates with decrease in the abundance of another. We used pairwise correlations reaching |r| >0.2 and permutation P < 0.05 to create correlation networks and algorithmically defined clusters of correlated species (Fig. 4a, Supplementary Fig. 4, Supplementary Data 7). We then mapped PD-associated species to the networks (Fig. 4b, Supplementary Fig. 4).

Species that showed the greatest increase in PD including *Bifidobacterium dentium*, *Actinomyces oris*, *Streptococcus mutans*, *Lactobacillus fermentum* and several other *Lactobacillus*, *Actinomyces*, and *Streptococcus* species were positively correlated with each other and mapped to cluster #2 of the PD network (Fig. 4). Cluster #2 included 41 species, 19 were associated with PD: 13 enriched, 6 reduced in PD, all connected via positive and negative correlations. Among them were 8 *Streptococcus* species: 4 elevated and 4 reduced in PD. Notably, *Streptococcus mutans* (elevated in PD) was negatively correlated with *Streptococcus* sp. *A12* (reduced in PD), which aligns with the original characterization of *Streptococcus* sp. *A12* as a novel strain that inhibits the growth and signaling pathways of the pathogenic *Streptococcus mutans*[26].

Most of the species that were depleted in PD, including short-chain fatty acid (SCFA) producing species of *Roseburia*, *Eubacterium*, *Ruminococcus*, and *Faecalibacterium prausnitzii* were positively correlated with each other and mapped to cluster #13 of the PD network (Fig. 4). Moreover, decreasing abundances of SCFA producing species correlated with increasing abundances of *Bifidobacterium* species (Fig. 4, Supplementary Data 5). These findings suggest competitive interactions occurring in the dysbiotic PD microbiome, both at large

**Table 1 | Subject characteristics and metadata**

| Metadata | | | PD | | NHC | | PD vs. NHC | |
|---|---|---|---|---|---|---|---|---|
| | | | *N* with data | Summary statistics | *N* with data | Summary statistics | *P* | OR [95%CI] |
| | | Number of subjects who passed QC | 490 | – | 234 | – | – | – |
| 1 | | Age[a] | 490 | 68.7 ± 8.5 | 234 | 65.8 ± 8.8 | 5E–5 | – |
| 2 | | Sex (N & % male)[a] | 490 | 310 (63%) | 234 | 70 (30%) | 3E–17 | 4 [2.9–5.7] |
| 3 | Ancestry | Hispanic or Latino | 477 | 9 (2%) | 224 | 3 (1%) | 0.76 | 1.4 [0.3–8.2] |
| 4 | | Race (N & % White) | 488 | 460 (94%) | 229 | 218 (95%) | 0.72 | 0.8 [0.4–1.8] |
| 5 | | Jewish | 457 | 6 (1%) | 223 | 3 (1%) | 1.00 | 1 [0.2–6.1] |
| 6 | Weight | BMI | 490 | 28 ± 5.4 | 230 | 28.4 ± 6.4 | 0.86 | – |
| 7 | | Lost >10 pounds in past year[b] | 479 | 133 (28%) | 228 | 36 (16%) | 5E–4 | 2 [1.3–3.2] |
| 8 | | Gained >10 pounds in past year | 472 | 88 (19%) | 228 | 40 (18%) | 0.76 | 1.1 [0.7–1.7] |
| 9 | Diet | Fruits or vegetables, daily | 478 | 270 (56%) | 229 | 152 (66%) | 0.01 | 0.7 [0.5–0.9] |
| 10 | | Poultry, beef, pork, seafood, or eggs, daily | 480 | 273 (57%) | 229 | 158 (69%) | 2E–3 | 0.6 [0.4–0.8] |
| 11 | | Nuts, daily | 479 | 83 (17%) | 229 | 54 (24%) | 0.05 | 0.7 [0.5–1] |
| 12 | | Yogurt ≥ few times a week | 479 | 98 (20%) | 229 | 68 (30%) | 8E–3 | 0.6 [0.4–0.9] |
| 13 | | Grains, daily | 480 | 274 (57%) | 229 | 143 (62%) | 0.19 | 0.8 [0.6–1.1] |
| 14 | | Alcohol[a] | 474 | 174 (37%) | 229 | 117 (51%) | 3E–4 | 0.6 [0.4–0.8] |
| 15 | | Cigarettes, cigars, pipe | 474 | 14 (3%) | 228 | 11 (5%) | 0.28 | 0.6 [0.2–1.5] |
| 16 | | Caffeine | 475 | 414 (87%) | 227 | 205 (90%) | 0.26 | 0.7 [0.4–1.2] |
| 17 | GI health day of stool collection | Constipation[b] | 428 | 99 (23%) | 211 | 12 (6%) | 7E–9 | 5 [2.6–10.2] |
| 18 | | Diarrhea | 437 | 8 (2%) | 221 | 5 (2%) | 0.77 | 0.8 [0.2–3.2] |
| 19 | | Abdominal pain[b] | 448 | 42 (9%) | 219 | 6 (3%) | 1E–3 | 3.7 [1.5–10.7] |
| 20 | | Excess gas[b] | 445 | 66 (15%) | 220 | 18 (8%) | 0.02 | 2 [1.1–3.6] |
| 21 | | Bloating[b] | 445 | 56 (13%) | 224 | 14 (6%) | 0.01 | 2.2 [1.2–4.3] |
| 22 | | Any of the above five GI items[b] | 438 | 170 (39%) | 203 | 38 (19%) | 3E–7 | 2.8 [1.8–4.2] |
| 23 | | Bristol stool chart[b] | 465 | 3.4 ± 1.5 | 222 | 3.9 ± 1.2 | 3E–6 | – |
| 24 | GI health past 3 months | Constipation[b] | 468 | 208 (44%) | 225 | 26 (12%) | 2E–19 | 6.1 [3.9–10] |
| 25 | | Diarrhea[b] | 464 | 35 (8%) | 223 | 28 (13%) | 0.05 | 0.6 [0.3–1] |
| 26 | GI disease | Colitis | 475 | 59 (12%) | 224 | 20 (9%) | 0.20 | 1.4 [0.8–2.6] |
| 27 | | Irritable bowel syndrome | 469 | 41 (9%) | 223 | 23 (10%) | 0.57 | 0.8 [0.5–1.5] |
| 28 | | Crohn's disease | 476 | 1 (0.2%) | 225 | 1 (0.4%) | 0.54 | 0.5 [0–37.2] |
| 29 | | Inflammatory bowel disease | 473 | 10 (2%) | 223 | 6 (3%) | 0.60 | 0.8 [0.3–2.7] |
| 30 | | Ulcers | 478 | 8 (2%) | 226 | 1 (0.4%) | 0.28 | 3.8 [0.5–170.6] |
| 31 | | Small intestine bacterial overgrowth | 466 | 1 (0.2%) | 224 | 0 (0%) | 1.00 | 1.4 [0.1–35.7] |
| 32 | | Celiac disease | 477 | 2 (0.4%) | 224 | 0 (0%) | 1.00 | 2.4 [0.1–49.4] |
| 33 | | GI cancer in the last 3 months | 464 | 1 (0.2%) | 219 | 0 (0%) | 1.00 | 1.4 [0.1–35] |
| 34 | | Any of the eight GI items | 444 | 107 (24%) | 220 | 43 (20%) | 0.20 | 1.3 [0.9–2] |
| 35 | Medications at time of stool collection unless noted | Indigestion | 471 | 172 (37%) | 224 | 77 (34%) | 0.61 | 1.1 [0.8–1.6] |
| 36 | | Antibiotics | 478 | 37 (8%) | 227 | 15 (7%) | 0.65 | 1.2 [0.6–2.4] |
| 37 | | Antibiotics in past 3 months | 471 | 116 (25%) | 225 | 59 (26%) | 0.64 | 0.9 [0.6–1.3] |
| 38 | | Laxatives[a] | 475 | 149 (31%) | 225 | 24 (11%) | 7E–10 | 3.8 [2.4–6.4] |
| 39 | | Anti-inflammatory | 469 | 158 (34%) | 221 | 66 (30%) | 0.34 | 1.2 [0.8–1.7] |
| 40 | | Probiotics[a] | 469 | 56 (12%) | 225 | 43 (19%) | 0.02 | 0.6 [0.4–0.9] |
| 41 | | Radiation or chemotherapy | 475 | 3 (1%) | 224 | 0 (0%) | 0.56 | 3.3 [0.2–64.7] |
| 42 | | Blood thinners | 473 | 116 (25%) | 224 | 44 (20%) | 0.18 | 1.3 [0.9–2] |
| 43 | | Cholesterol | 476 | 202 (42%) | 224 | 97 (43%) | 0.87 | 1 [0.7–1.3] |
| 44 | | Blood pressure | 476 | 234 (49%) | 225 | 119 (53%) | 0.37 | 0.9 [0.6–1.2] |
| 45 | | Thyroid | 476 | 82 (17%) | 225 | 45 (20%) | 0.40 | 0.8 [0.5–1.3] |
| 46 | | Asthma or COPD | 476 | 35 (7%) | 223 | 17 (8%) | 0.88 | 1 [0.5–1.9] |
| 47 | | Diabetes | 473 | 64 (14%) | 224 | 35 (16%) | 0.49 | 0.8 [0.5–1.4] |
| 48 | | Pain[a] | 474 | 110 (23%) | 224 | 36 (16%) | 0.04 | 1.6 [1–2.5] |
| 49 | | Depression, anxiety, mood[a] | 477 | 180 (38%) | 224 | 51 (23%) | 7E–5 | 2.1 [1.4–3] |
| 50 | | Birth control or estrogen | 175 | 9 (5%) | 157 | 19 (12%) | 0.03 | 0.4 [0.2–1] |

## Table 1 (continued) | Subject characteristics and metadata

| Metadata | | PD | | NHC | | PD vs. NHC | |
|---|---|---|---|---|---|---|---|
| | | *N* with data | Summary statistics | *N* with data | Summary statistics | *P* | OR [95%CI] |
| 51 | Antihistamines[a] | 477 | 80 (17%) | 224 | 73 (33%) | 5E–6 | 0.4 [0.3–0.6] |
| 52 | Co-Q 10 | 477 | 61 (13%) | 225 | 32 (14%) | 0.63 | 0.9 [0.5–1.5] |
| 53 | Sleep aid[a] | 470 | 196 (42%) | 223 | 58 (26%) | 7E–5 | 2 [1.4–2.9] |

In total, 490 PD and 234 neurologically healthy control (NHC) subjects completed the enrollment process and were studied (see STORMs flowchart in Fig. 1). Metadata were collected using self-administered questionnaires (questionnaires are provided in Supplementary Fig. 1). Here, we show the data on 53 variables relevant to gut microbiome. N with data: number of subjects for whom data on the specified variable was available. Summary statistics: distribution of the variable in PD and NHC, measured as mean ± SD for continuous variables, or the numbers and percentages for dichotomous variables. The frequency/distribution of each variable was tested for difference between PD and NHC. *P*: uncorrected, two-sided *P*-value of the difference between PD and NHC (more conservative than multiple testing corrected P because the aim is to identify potential confounders) derived from Fisher's exact test if variable was categorical, or Wilcoxon rank-sum test if quantitative. OR [95%CI]: odds ratio and 95% confidence interval of the difference between PD and NHC.

[a,b]Variables that differed significantly in PD vs NHC.
[a]Variables whose effects on association of species with PD were investigated by adding them to analysis as covariates.
[b]Intrinsic to PD (constipation, GI problems and weight loss are common features of PD); they were not included as covariates because adjusting for them would have masked a part of PD. Food groups were not tagged because, although several reached significances, they were all lower in PD, which suggests PD subjects eat less frequently of all food groups. Taking birth control or estrogen was also not tagged because it was exclusive to females and uncommon.

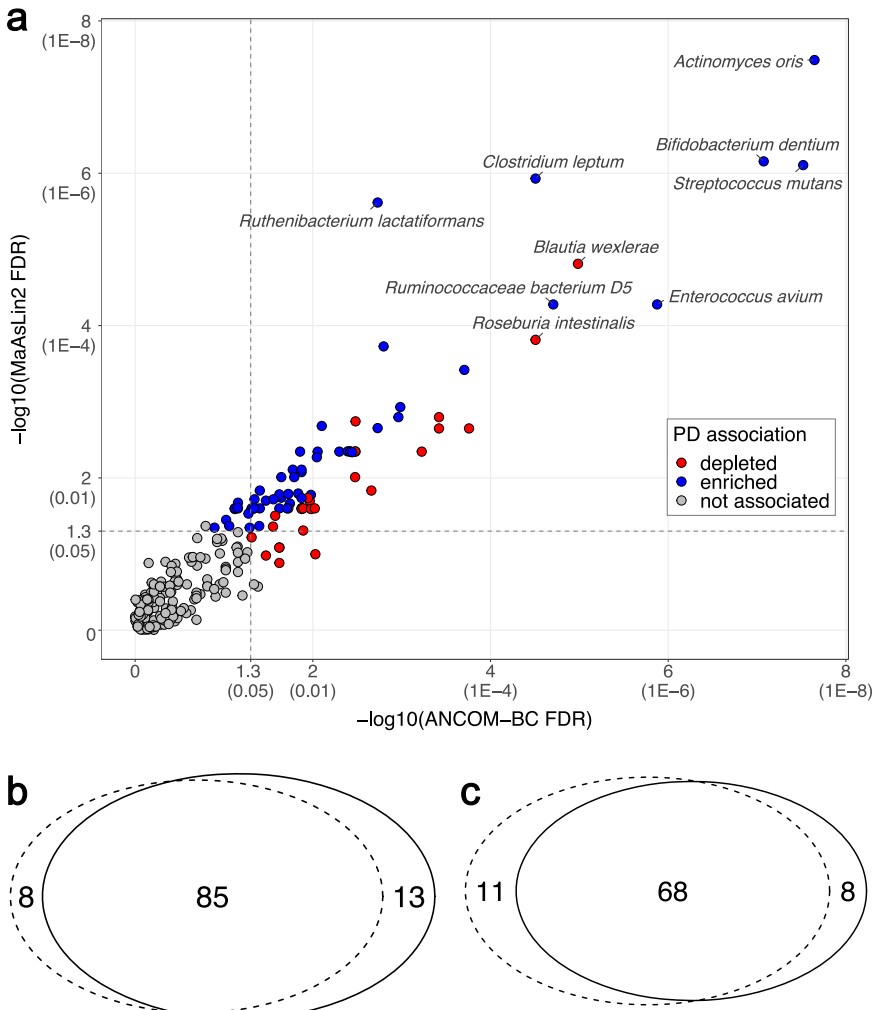

**Fig. 2 | PD-associated species nominated by consensus of MaAsLin2 and ANCOM-BC.** Analysis included *N* = 724 biologically independent samples from 490 PD and 234 neurologically healthy control (NHC) subjects. **a** 257 species (denoted by circles in the plot) were tested in microbiome-wide association study (MWAS) with two statistical methods: MaAsLin2 and ANCOM-BC. The results are shown according to significance (−log10 of the FDR) achieved by MaAsLin2 (*Y*-axis) vs ANCOM-BC (*X*-axis). Corresponding untransformed FDR values are provided in parentheses on the *X* and *Y* axes for easier interpretation. 84 species were nominated as PD-associated, defined by FDR < 0.05 by one method and FDR≤0.1 by the

other: 68 achieved FDR < 0.05 by both methods, 10 achieved ANCOM-BC MaAsLin2 FDR < 0.05 by MaAsLin2 and FDR≤0.1 by ANCOM-BC, and 6 achieved FDR < 0.05 by ANCOM-BC and FDR≤0.1 by MaAsLin2. Blue: abundance is significantly elevated in PD. Red: abundance is significantly reduced in PD. Gray: not significantly associated with PD. Vertical and horizontal dashed lines denote points on *X* and *Y* axes that correspond to FDR=0.05. **b, c** Venn diagrams summarizing the overlap of species detected by MaAsLin2 (dotted circle) and ANCOM-BC (solid circle) at FDR≤0.1 (**b**), and at FDR < 0.05 (**c**).

**Table 2 | Identification and characterization of 84 PD-associated species**

| (i) PD-associated species | (ii) Prevalence | | (iii) MWAS via MaAsLin2 | | | (iv) MWAS via ANCOM-BC | | | (v) Cluster # | (vi) Analysis of confounding |
|---|---|---|---|---|---|---|---|---|---|---|
| | PD | NHC | FDR | FC | RA in NHC | FDR | FC | BC-OA in NHC | | |
| *Acidaminococcus intestini* | 157 | 51 | 0.02 | 2.81 | 0.50% | 0.02 | 3.26 | 13 | 11 | PD+ |
| *Actinomyces naeslundii* | 77 | 19 | 0.02 | 1.41 | <0.01% | 0.05 | 1.57 | 2 | 2 | PD++ |
| *Actinomyces oris* | 269 | 76 | 3E-8 | 3.54 | 0.01% | 2E-8 | 6.52 | 11 | 2 | PD++, A-, P+, M+ |
| *Actinomyces sp HPA0247* | 126 | 31 | 5E-3 | 1.78 | 0.01% | 5E-3 | 2.28 | 3 | 2 | PD++ |
| *Actinomyces sp ICM47* | 132 | 84 | 0.02 | 0.60 | 0.01% | 0.01 | 0.40 | 18 | 2 | L-- |
| *Actinomyces sp oral taxon 448* | 55 | 6 | 4E-3 | 1.47 | <0.01% | 4E-3 | 1.63 | 1 | 2 | PD++ |
| *Alistipes finegoldii* | 399 | 175 | 0.04 | 2.14 | 0.91% | 0.09 | 2.17 | 2666 | 13 | PD++, M- |
| *Alistipes indistinctus* | 224 | 82 | 8E-3 | 2.50 | 0.06% | 0.02 | 2.93 | 27 | 10 | PD++, A+, P- |
| *Anaerostipes hadrus* | 337 | 187 | 2E-3 | 0.32 | 1.12% | 4E-4 | 0.19 | 11565 | 13 | PD--, A+, M- |
| *Bacteroides faecis CAG 32* | 77 | 19 | 0.04 | 1.50 | 0.08% | 0.04 | 1.92 | 2 | 5 | PD++, A++ |
| *Bifidobacterium bifidum* | 153 | 54 | 0.03 | 2.48 | 0.37% | 0.04 | 2.64 | 10 | 8 | PD++, Pr++ |
| *Bifidobacterium breve* | 78 | 26 | 0.05 | 1.59 | 0.05% | 0.13 | 1.70 | 3 | 8 | L+, P+ |
| *Bifidobacterium dentium* | 195 | 48 | 7E-7 | 4.08 | 0.09% | 8E-8 | 7.06 | 6 | 2 | PD++, A- |
| *Bifidobacterium gallinarum* | 40 | 5 | 0.02 | 1.26 | <0.01% | 0.02 | 1.45 | 1 | 8 | L++ |
| *Bifidobacterium longum* | 402 | 173 | 0.03 | 2.34 | 1.53% | 0.05 | 2.74 | 3887 | 8 | A-, S+ |
| *Bifidobacterium pullorum* | 53 | 11 | 0.02 | 1.49 | 0.01% | 0.02 | 1.76 | 2 | 8 | |
| *Bifidobacterium saeculare* | 35 | 4 | 0.02 | 1.27 | <0.01% | 0.03 | 1.38 | 1 | 8 | |
| *Blautia hansenii* | 78 | 59 | 0.02 | 0.53 | 0.07% | 0.01 | 0.35 | 11 | 10 | PD-- |
| *Blautia wexlerae* | 433 | 214 | 2E-5 | 0.26 | 2.83% | 1E-5 | 0.20 | 53145 | 8 | PD-- |
| *Butyricicoccus pullicaecorum* | 22 | 21 | 0.03 | 0.76 | 0.02% | 0.03 | 0.57 | 2 | none | PD- |
| *C. Methanomassiliicoccus intestinalis* | 45 | 9 | 0.03 | 1.34 | 0.01% | 0.05 | 1.58 | 2 | 10 | PD++ |
| *Christensenella minuta* | 76 | 23 | 0.03 | 1.47 | <0.01% | 0.08 | 1.51 | 2 | 10 | L+ |
| *Cloacibacillus evryensis* | 102 | 29 | 8E-3 | 1.68 | 0.03% | 0.01 | 2.24 | 3 | 10 | PD++, M+ |
| *Clostridium hylemonae* | 34 | 5 | 0.02 | 1.23 | <0.01% | 0.07 | 1.31 | 1 | none | PD++ |
| *Clostridium innocuum* | 373 | 160 | 0.02 | 1.97 | 0.16% | 0.07 | 2.21 | 437 | 10 | |
| *Clostridium leptum* | 432 | 174 | 1E-6 | 3.51 | 0.13% | 3E-5 | 4.41 | 919 | 8 | PD++ |
| *Clostridium sp CAG 299* | 68 | 52 | 2E-3 | 0.46 | 0.16% | 3E-3 | 0.29 | 11 | 10 | PD-- |
| *Clostridium sp CAG 58* | 305 | 157 | 0.06 | 0.54 | 0.35% | 0.05 | 0.37 | 1555 | 13 | L--, M- |
| *Collinsella massiliensis* | 261 | 96 | 2E-3 | 1.84 | <0.01% | 8E-3 | 2.20 | 13 | 11 | PD++, A+, P++ |
| *Coprobacillus cateniformis* | 82 | 12 | 4E-4 | 1.90 | <0.01% | 2E-4 | 2.37 | 2 | 10 | PD++, L++ |
| *Dialister invisus* | 112 | 72 | 0.03 | 0.57 | 0.04% | 0.02 | 0.36 | 19 | 9 | PD-- |
| *Eisenbergiella tayi* | 266 | 100 | 2E-4 | 2.69 | 0.03% | 2E-3 | 3.61 | 33 | 10 | PD++, L++, M+ |
| *Enorma massiliensis* | 128 | 38 | 0.03 | 1.60 | 0.04% | 0.05 | 1.69 | 3 | 11 | PD++, A+ |
| *Enterococcus avium* | 53 | 1 | 5E-5 | 1.76 | <0.01% | 1E-6 | 2.60 | 1 | 8 | PD++ |
| *Enterococcus faecium* | 63 | 11 | 0.03 | 1.40 | 0.05% | 0.02 | 1.85 | 2 | 8 | |
| *Escherichia coli* | 285 | 105 | 5E-3 | 2.47 | 0.87% | 9E-3 | 4.03 | 105 | 6 | PD++ |
| *Eubacterium callanderi* | 79 | 20 | 0.03 | 1.44 | 0.01% | 0.05 | 1.73 | 2 | 13 | L+ |
| *Eubacterium eligens* | 234 | 132 | 0.10 | 0.56 | 0.50% | 0.03 | 0.32 | 569 | 13 | A++, M-- |
| *Eubacterium hallii* | 331 | 176 | 0.08 | 0.59 | 0.72% | 0.02 | 0.33 | 5441 | 13 | A+ |
| *Eubacterium limosum* | 81 | 23 | 0.01 | 1.59 | <0.01% | 0.04 | 1.67 | 2 | none | PD++ |
| *Eubacterium ramulus* | 159 | 94 | 0.04 | 0.53 | 0.15% | 0.03 | 0.35 | 62 | 13 | |
| *Eubacterium rectale* | 390 | 206 | 0.01 | 0.36 | 4.04% | 2E-3 | 0.24 | 28407 | 13 | PD- |
| *Eubacterium sp CAG 38* | 189 | 117 | 0.01 | 0.41 | 0.20% | 3E-3 | 0.24 | 163 | 13 | L- |
| *Faecalibacterium prausnitzii* | 447 | 216 | 5E-3 | 0.40 | 5.62% | 3E-3 | 0.33 | 150387 | 13 | PD-, A++, P-, M-- |
| *Fusicatenibacter saccharivorans* | 350 | 192 | 5E-3 | 0.28 | 2.15% | 6E-4 | 0.19 | 24835 | 13 | PD-, A+, L-, M- |
| *Harryflintia acetispora* | 143 | 47 | 0.04 | 1.50 | <0.01% | 0.09 | 1.69 | 5 | 13 | PD++ |
| *Hungatella hathewayi* | 264 | 98 | 5E-3 | 2.31 | 0.04% | 0.01 | 2.90 | 32 | 10 | PD+, A-, L++ |
| *Klebsiella pneumoniae* | 143 | 47 | 0.02 | 1.97 | 0.55% | 0.03 | 2.73 | 8 | 6 | PD+, A- |
| *Klebsiella quasipneumoniae* | 101 | 26 | 0.03 | 1.49 | 0.05% | 0.02 | 2.27 | 3 | 6 | PD+, A-- |
| *Lachnoclostridium sp An131* | 73 | 17 | 0.01 | 1.43 | <0.01% | 0.02 | 1.64 | 2 | 7 | PD+, L++ |
| *Lactobacillus fermentum* | 113 | 28 | 4E-3 | 2.18 | 0.05% | 4E-3 | 2.70 | 3 | 2 | PD+, A-, P++ |
| *Lactobacillus gasseri* | 50 | 6 | 0.02 | 1.33 | <0.01% | 0.01 | 1.68 | 1 | 2 | A-, P+ |

**Table 2 (continued) | Identification and characterization of 84 PD-associated species**

| (i) PD-associated species | (ii) Prevalence | | (iii) MWAS via MaAsLin2 | | | (iv) MWAS via ANCOM-BC | | | (v) Cluster # | (vi) Analysis of confounding |
|---|---|---|---|---|---|---|---|---|---|---|
| | PD | NHC | FDR | FC | RA in NHC | FDR | FC | BC-OA in NHC | | |
| *Lactobacillus paragasseri* | 51 | 6 | 5E-3 | 1.58 | 0.01% | 4E-3 | 1.82 | 1 | 2 | PD++ |
| *Lactobacillus reuteri* | 42 | 8 | 0.03 | 1.57 | 0.04% | 0.05 | 1.57 | 1 | 8 | PD+ |
| *Lactobacillus rhamnosus* | 105 | 29 | 0.01 | 1.73 | 0.01% | 0.02 | 2.15 | 3 | 8 | PD+, A–, P+ |
| *Lactobacillus salivarius* | 71 | 13 | 2E-3 | 1.88 | 0.01% | 1E-3 | 2.31 | 2 | 2 | PD++ |
| *Megasphaera sp DISK 18* | 52 | 7 | 5E-3 | 1.66 | 0.01% | 4E-3 | 1.87 | 1 | 11 | PD++, L++ |
| *Megasphaera sp MJR8396C* | 83 | 16 | 0.02 | 1.88 | 0.10% | 0.01 | 2.24 | 2 | 11 | PD+ |
| *Methanobrevibacter smithii* | 184 | 63 | 8E-3 | 2.63 | 0.25% | 0.01 | 3.26 | 19 | 10 | PD++, A+ |
| *Monoglobus pectinilyticus* | 178 | 109 | 0.08 | 0.54 | 0.18% | 0.02 | 0.32 | 125 | 1 | L– |
| *Parabacteroides distasonis* | 435 | 195 | 0.03 | 2.16 | 1.98% | 0.08 | 2.19 | 19641 | 1 | PD++ |
| *Porphyromonas asaccharolytica* | 51 | 8 | 5E-3 | 1.75 | <0.01% | 9E-3 | 1.64 | 1 | 17 | PD++ |
| *Prevotella copri* | 157 | 103 | 0.03 | 0.44 | 3.52% | 0.01 | 0.27 | 66 | 10 | PD–, A+ |
| *Pseudoflavonifractor sp An184* | 69 | 19 | 0.03 | 1.50 | 0.01% | 0.07 | 1.60 | 2 | 7 | L++ |
| *Roseburia faecis* | 290 | 158 | 0.03 | 0.37 | 1.77% | 0.01 | 0.25 | 2513 | 13 | |
| *Roseburia intestinalis* | 294 | 170 | 2E-4 | 0.21 | 1.02% | 3E-5 | 0.13 | 3731 | 13 | PD––, M– |
| *Roseburia inulinivorans* | 270 | 151 | 0.03 | 0.44 | 0.40% | 0.01 | 0.28 | 921 | 13 | A++, L– |
| *Rothia mucilaginosa* | 150 | 91 | 0.03 | 0.59 | 0.02% | 0.01 | 0.41 | 22 | 2 | PD–– |
| *Ruminococcaceae bacterium D5* | 161 | 38 | 5E-5 | 2.36 | 0.02% | 2E-5 | 4.18 | 4 | 10 | PD++, L–– |
| *Ruminococcus bicirculans* | 146 | 95 | 0.03 | 0.42 | 1.12% | 9E-3 | 0.23 | 131 | 13 | L–, P–, M–– |
| *Ruminococcus callidus* | 17 | 22 | 4E-3 | 0.81 | 0.01% | 3E-3 | 0.49 | 2 | none | PD–– |
| *Ruminococcus lactaris* | 115 | 88 | 2E-3 | 0.43 | 0.35% | 4E-4 | 0.17 | 69 | 13 | PD––, P– |
| *Ruthenibacterium lactatiformans* | 479 | 228 | 2E-6 | 2.17 | 0.50% | 2E-3 | 1.89 | 28593 | 10 | PD++, L++, M+ |
| *Scardovia wiggsiae* | 50 | 5 | 2E-3 | 1.48 | <0.01% | 2E-3 | 1.65 | 1 | 2 | PD++, P++, M+ |
| *Streptococcus anginosus* group | 111 | 30 | 0.02 | 1.53 | 0.01% | 0.01 | 2.06 | 3 | 2 | PD+ |
| *Streptococcus australis* | 40 | 46 | 2E-3 | 0.66 | 0.01% | 2E-4 | 0.37 | 5 | 2 | PD–– |
| *Streptococcus infantis* | 27 | 23 | 0.13 | 0.84 | <0.01% | 0.02 | 0.64 | 2 | 2 | PD– |
| *Streptococcus lutetiensis* | 36 | 5 | 0.05 | 1.33 | 0.07% | 0.05 | 1.56 | 1 | 8 | PD++ |
| *Streptococcus mitis* | 67 | 48 | 0.02 | 0.68 | 0.01% | 0.01 | 0.48 | 5 | 2 | PD––, L–– |
| *Streptococcus mutans* | 171 | 35 | 8E-7 | 3.11 | 0.04% | 3E-8 | 5.76 | 4 | 2 | PD++ |
| *Streptococcus sp A12* | 31 | 27 | 0.05 | 0.82 | <0.01% | 0.01 | 0.59 | 2 | 2 | PD– |
| *Streptococcus vestibularis* | 185 | 60 | 0.02 | 1.65 | 0.06% | 0.02 | 2.61 | 11 | 2 | PD++ |
| *Turicibacter sanguinis* | 105 | 24 | 1E-3 | 2.14 | 0.01% | 1E-3 | 2.57 | 2 | 12 | PD++, A–, M–– |
| *Veillonella dispar* | 46 | 41 | 0.10 | 0.83 | 0.01% | 9E-3 | 0.47 | 4 | 12 | |

Metagenome-wide association study (MWAS) was conducted to test differential abundances of microbial species in 490 persons with PD vs. 234 neurologically healthy controls (NHC). While 719 species were detected and passed QC, 462 were excluded from MWAS because they were present in <5% of subjects; hence MWAS included 257 species that were present in ≥5% of subjects (minimum analytic N = 37). Two statistical methods were used: MaAsLin2 and ANCOM-BC. In both we adjusted for stool sample collection method and total sequence count per sample. (i) 84 species emerged from MWAS as PD-associated, based on reaching significance by both MaAsLin2 and ANCOM-BC (i.e., FDR < 0.05 by one method and FDR≤0.1 in the other). (ii) Prevalence is the number of subjects in whom the species was detected (out of 490 PD, and 234 NHC). (iii and iv) For each PD-associated species, we show the results from MaAsLin2 and ANCOM-BC, including FDR for statistical significance of association, the fold change (FC) in abundance of species in PD vs. NHC, and the abundance of species in NHC, which is relative abundance (RA) for MaAsLin2, and sampling bias-corrected observed abundance (BC-OA) for ANCOM-BC. Full species-level MWAS results, including standard errors, are provided in Supplementary Data 1. (v) Network analysis revealed that PD-associated species are not independent. Their abundances are correlated (Supplementary Data 5). They form polymicrobial clusters that tend to increase or decrease in abundances together. The cluster # assignment is shown here to enable easy identification of the species that are correlated with each other. For example, *Escherichia coli* and *Klebsiella* spp. are increased in PD and are positively correlated (cluster #6). (vi) Among the metadata analyzed (Table 1), seven variables were identified as potential confounders. The 84 species were re-tested with MaAsLin2 in an extended model that included case status, total sequence count per sample, and alcohol (A), laxative (L), pain medication (P), depression/anxiety/mood medication (M), probiotics (Pr), antihistamine, and sleep aid (S) use. The confounder analysis included 435 PD and 219 NHC for whom data on all variables were available. We show the variables that had a signal for association with the species, and if PD is listed, the association of species with PD persisted through adjusting for all the variables. The direction of association is noted with + or –. The statistical significance is noted by the number of + or – (+ +/–– is FDR < 0.05 and +/– is FDR < 0.15). Note that the inclusion of 9 variables in the model severely diminishes statistical power, to the extent that 7 of the 84 species completely lost their signal (empty cells). Full results of confounding analysis are shown in Supplementary Data 4.

community scale (*Bifidobacterium* species vs. SCFA-producing species), as well as at species scale (*Streptococcus* sp. *A12* vs. *Streptococcus mutans*).

We noted two clusters with species that are known to cause infections. These species are commensal to gut microbiome, as evinced by their presence in NHC, but have the capacity to become opportunistic pathogens. *Escherichia coli*, *Klebsiella pneumoniae* and *Klebsiella quasipneumoniae*, which were elevated in PD, were positively correlated and grouped into cluster #6 of the PD network. Cluster #17 is a polymicrobial community of 19 opportunistic pathogen species (Supplementary Data 8). We detected

overabundance of these taxa at genus-level in our prior datasets[17]. Here, at species-level, only *Porphyromonas asaccharolytica* was prevalent enough to be included in MWAS and it confirmed to be significantly elevated in PD. The other 18 species were too rare individually to be tested in MWAS. When taken together, the relative abundance of species in cluster #17 was significantly elevated in PD (fold change (FC) = 2.63, 95% CI = 1.7–4.1, P = 2E-5). Not only the prevalence and abundance of these species were elevated in PD, but they also formed a tightly interconnected polymicrobial cluster in PD metagenome, as they do in clinical infection specimen[27], but not in NHC metagenome. In PD, each species in cluster #17 connects with an

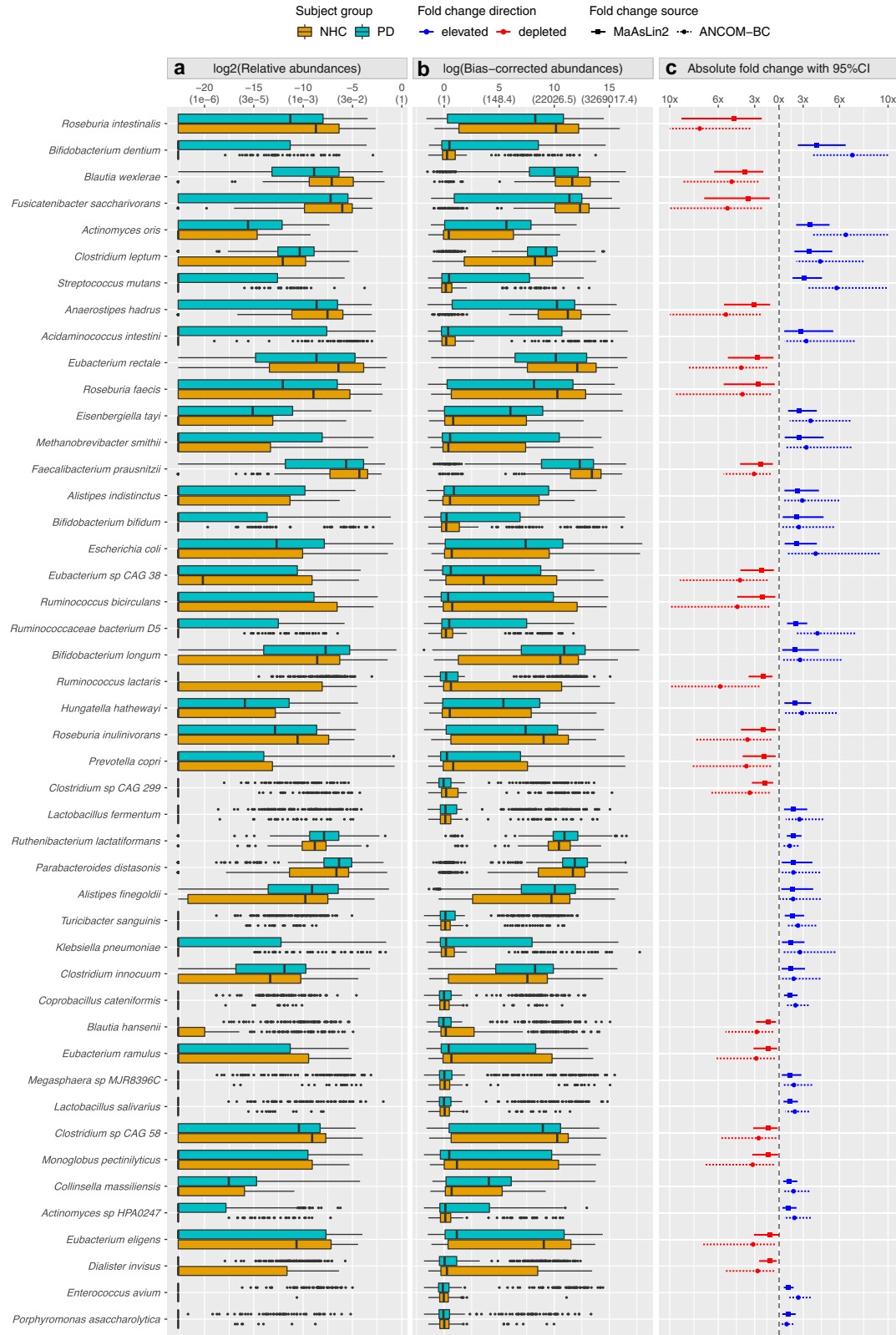

average of 5.4 other species, as compared to 1 in NHC (Supplementary Data 8).

### Replication

We have previously characterized two cohorts of PD and NHC subjects using 16S sequencing ($N1$ = 333 and $N2$ = 507)[17,28]. Except for 11 subjects who were inadvertently double-enrolled, the 724 subjects in this shotgun study are independent of the 840 subjects in the 16S studies. Having these three large datasets presented a unique opportunity to compare 16S and shotgun results, and to replicate and validate the findings. In prior 16S studies[17,28], we reported association of 15 genera with PD and showed that they form three clusters of (1) opportunistic pathogens (elevated in PD), (2) SCFA-producing bacteria (reduced), and (3) *Bifidobacterium* and *Lactobacillus* (elevated). Here, we readily

**Fig. 3 | Differential abundances and effect sizes of PD-associated species.** Analysis included $N$ = 724 biologically independent samples from 490 PD and 234 neurologically healthy control (NHC) subjects. Forty-six species that had at least 75% (and up to 750%) change in abundance in PD are shown here; for all 84 PD-associated species see Supplementary Fig. 3. **a** Distribution of relative abundances. Log2 transformed relative abundance values, as used in MaAsLin2, were used to generate the boxplots. Untransformed relative abundances, shown in parenthesis, are provided on the $X$-axis for easier interpretation of data. Boxplots show distribution of the data for PD (blue green) and NHC (orange). Each sample was plotted according to its abundance of the species. The left, middle, and right vertical boundaries of each box represents the first, second (median), and third quartiles of the data; that is, 25% of samples have abundance lower than the left border of the box, 25% of samples have abundances that are higher than the right border of the box. Absence of a box indicates 75% of samples had zero abundance.

The lines extending from the two ends of each box represent 1.5x outside the interquartile range (range = (abundance value at 75% minus abundance value at 25%) x 1.5). Points beyond the lines are outlier samples. **b** Distribution of bias-corrected observed abundances (used in ANCOM-BC). Natural log transformed sampling-bias corrected observed abundances, as used in ANCOM-BC, were used here to generate the boxplots. Untransformed bias-corrected observed abundances, shown in parenthesis, are provided on the $X$-axis for easier interpretation of data. Description of boxplots are the same as for (**a**). **c** Absolute fold change in differential abundance in PD vs. NHC (squares and circles) and its 95% confidence interval (CI; solid and dashed lines), calculated from beta and standard errors estimated by MaAsLin2 (square with solid line of 95% CI), and ANCOM-BC (circle with dotted line of 95% CI). CI are truncated at 10x. Points and lines for fold changes and 95% CI were colored blue (elevated in PD) or red (reduced in PD).

confirmed 13 of the 15 genus-level associations with statistical significance (Table 3). The remaining two (*Oscillospira*, and *Prevotella* sub-genus) were not captured here due to differences in reference databases used for taxonomic assignment: SILVA v132 for 16S and ChocoPhlAn v30 for shotgun. *Oscillospira*, a genus designation in SILVA, was not in ChocoPhlAn. *Prevotella*, as called by SILVA, was in fact a sub-genus of *Prevotella* and included species *P. buccalis*, *P. timonensis*, *P. disiens*, *P. bivia*, *P. amnii*, and *P. oralis*. Although detected, none of these species was tested because they were individually rare. We grouped them to recreate the *Prevotella* sub-genus, and successfully replicated the positive association with PD (FC = 1.5, 95% CI = 1.2–1.9, P = 1E-3). We also captured and replicated the previously defined genus clusters of opportunistic pathogens (cluster #17 here), SCFA-producing bacteria (cluster #13), and *Bifidobacterium* and *Lactobacillus* (cluster #2). Thus, using metagenomics, we replicated both MWAS and network analysis findings of 16S studies in an independent dataset, and resolved them to species-level.

**Results align with existing literature and resolve inconsistencies**
Prior studies of PD and microbiome that were conducted using 16S all detected a significant dysbiosis in PD gut, but results on PD-associated taxa were widely inconsistent. Small sample sizes and inter-study variations were thought responsible. Recent meta-analyses were able to identify few associations at family or genus level robust enough to transcend inter-study variations[18–20]. We successfully confirmed these associations at genus level and resolved them to species, including *Blautia*, *Faecalibacterium*, *Fusicatenibacter*, *Roseburia* and *Ruminococcus*, which are reduced in PD, and *Bifidobacterium*, *Hungatella*, *Lactobacillus*, *Methanobrevibacter* and *Porphyromonas*, which are elevated in PD[18–20]. *Prevotella* has been reported as decreased in PD by some[29,30] and increased by others[17,28]. At species-level, *Prevotella copri* was decreased, and the pathogenic species of *Prevotella* (as defined by SILVA above) were increased as a group, confirming, and resolving the seemingly contradictory reports on *Prevotella*. *Akkermansia* is a conundrum. Most studies report elevated *Akkermansia* in PD. We did not detect a statistically significant signal for *Akkermansia* at genus or species level in this southern US cohort, although the trend was elevated. Interestingly, *Akkermansia* was elevated in our two prior 16S datasets as well, but reached significance only in the multi-state cohort that was primarily from northern US[28] and not the dataset from southern US[17], suggesting a geographic effect. It is not known which of these changes are specific to PD. Elevated *Bifidobacterium* and *Lactobacillus* and loss of SCFA producing bacteria have also been observed in other inflammatory disorders of the gut[31].

Analyses conducted at genus or higher taxonomic levels have the underlying assumption that microorganisms within each classification have similar trend of association with disease and can therefore be collapsed. If the assumption does not hold, signals may be missed or be inconsistent across studies (as was shown for *Prevotella* above). We encountered several such heterogenous genera (Table 3). Most notably, we detected 8 PD-associated *Streptococcus* species: 4 were elevated in PD, 4 were reduced. However, at genus-level, *Streptococcus* lacked evidence for association with PD. Thus, *Streptococcus*, the genus with the greatest number of PD-associated species, was missed at genus-level because of heterogeneity.

Only two prior PD studies used shotgun sequencing, conducted by Bedarf et al.[21] and Qian et al.[22]. Compared to Bedarf et al.[21] we differed in sample size ($N$ = 59 vs. $N$ = 724), sex (100% male vs. 52% male), patient characteristics such as constipation (not relevant vs. highly significant), geography (Bonn, EU vs. Deep South, US), and taxonomic profiling (MOCAT2 vs. MetaPhlAn3), to name a few of a myriad of factors that can have profound effects on results. We confirmed their original report of reduction in *Prevotella copri* and alterations in tryptophan metabolism. They also reported reduction in total virus abundance in PD. We have not yet investigated the virome because virus databases are still sparse and detection methods underperform[32]. Qian et al.'s study[22] also differed from our study in many aspects, most notably in race (Chinese vs. Caucasian) and study design (gene-based machine learning classifier for prediction vs. delineation of alterations at global, species, gene, and pathway level and functional inference). Despite differences, we confirmed their report that majority of species enriched in PD were in *Firmicutes* phylum, and *Alistipes* of *Bacteroidetes* phylum was also highly elevated in PD.

**Species, genus, and cluster level analyses are complementary**
Species-level testing can capture the signals that are lost at genus level due heterogeneity within genus. In fact, we detected associations with 22 species belonging to 12 genera where the associations were missed at genus-level MWAS (Table 3). Conversely, when species are rare and excluded from testing for statistical reasons, the signal may be present at genus level due to cumulative effects of species with similar effects (e.g., *Corynebacterium*). Networks of correlating species is another complimentary source of information as it may reveal disease-relevant polymicrobial clusters. Consider cluster #17 composed of 19 opportunistic pathogen species from 10 genera: species-level MWAS detected only one of the 19 species as elevated in PD (*Porphyromonas asaccharolytica*), genus-level MWAS detected 3 of the 10 genera as elevated in PD (*Actinomyces*, *Corynebacterium*, *Porphyromonas*), and cluster-level analysis detected the whole polymicrobial cluster as elevated in PD.

**Altered abundances of microbial gene families and metabolic pathways**
We identified 8,528 gene families (KEGG ortholog (KO) groups) and 511 metabolic pathways. Excluding those present in <5% of subjects, 4,679 KO groups and 407 metabolic pathways were tested for differential abundance in PD vs. NHC using MaAsLin2 and ANCOM-BC. The full results are provided for KO groups (Supplementary Data 9) and pathways (Supplementary Data 10). According to MaAsLin2, 50% of KO groups and 67% of pathways were affected in PD. According to ANCOM-BC, 32% of KO groups and 55% of pathways were affected. The

**Table 3 | PD-associated genera, replication of prior findings, and evidence of heterogeneity**

| (i) PD-associated genera | (ii) Prevalence | | (iii) MWAS via MaAsLin2 | | | (iv) MWAS via ANCOM-BC | | | (v) Replication | (vi) Heterogeneity | | | |
|---|---|---|---|---|---|---|---|---|---|---|---|---|---|
| | PD | NHC | FDR | FC | RA in NHC | FDR | FC | BC-OA in NHC | (Reference) | N spp. tested | N ↑ in PD | N ↓ in PD | Genus signal |
| Acidaminococcus | 158 | 51 | 9E–3 | 2.89 | 0.50% | 0.02 | 3.22 | 13 | 18, 20 | 1 | 1 | 0 | ↑ |
| Actinomyces | 410 | 166 | 2E–3 | 2.08 | 0.05% | 7E–3 | 2.66 | 446 | | 10 | 4 | 1 | ↑ |
| Anaerostipes | 399 | 210 | 4E–4 | 0.35 | 1.15% | 1E–5 | 0.21 | 31470 | 19 | 2 | 0 | 1 | ↓ |
| Bifidobacterium | 453 | 208 | 4E–3 | 2.53 | 3.98% | 0.04 | 2.40 | 40087 | 17–19 | 10 | 7 | 0 | ↑ |
| Blautia | 487 | 234 | 5E–6 | 0.63 | 7.42% | 2E–8 | 0.50 | 925208 | 17, 19 | 11 | 0 | 2 | ↓ |
| Butyricicoccus | 25 | 22 | 0.06 | 0.78 | 0.02% | 0.02 | 0.56 | 2 | 17 | 1 | 0 | 1 | ↓ |
| Christensenella | 76 | 23 | 0.02 | 1.47 | <0.01% | 0.13 | 1.44 | 2 | 18, 19 | 1 | 1 | 0 | ↑ |
| Cloacibacillus | 123 | 35 | 3E–3 | 1.92 | 0.04% | 7E–3 | 2.53 | 4 | 18, 19 | 2 | 1 | 0 | ↑ |
| Clostridium | 413 | 207 | 3E–3 | 0.46 | 0.79% | 2E–3 | 0.32 | 20444 | | 7 | 0 | 2 | ↓ |
| Coprobacillus | 82 | 12 | 3E–4 | 1.90 | <0.01% | 4E–4 | 2.26 | 2 | 20 | 1 | 1 | 0 | ↑ |
| Corynebacterium | 38 | 6 | 0.02 | 1.31 | <0.01% | 0.13 | 1.30 | 1 | 17 | 0 | 0 | 0 | ↑ |
| Eisenbergiella | 382 | 161 | 1E–3 | 2.80 | 0.19% | 0.01 | 2.96 | 575 | | 2 | 1 | 0 | ↑ |
| Enorma | 264 | 99 | 3E–3 | 1.96 | 0.04% | 0.01 | 2.19 | 16 | | 2 | 2 | 0 | ↑ |
| Enterococcus | 121 | 24 | 1E–4 | 2.28 | 0.06% | 6E–5 | 3.98 | 3 | | 3 | 2 | 0 | ↑ |
| Erysipelatoclostridium | 400 | 169 | 9E–3 | 2.11 | 0.20% | 0.05 | 2.37 | 799 | | 3 | 1 | 0 | ↑ |
| Escherichia | 285 | 105 | 3E–3 | 2.47 | 0.87% | 8E–3 | 3.86 | 104 | 18 | 1 | 1 | 0 | ↑ |
| Eubacterium | 447 | 222 | 0.04 | 0.58 | 2.21% | 2E–3 | 0.41 | 126275 | 18, 19 | 11 | 2 | 4 | ↓ |
| Faecalibacterium | 447 | 216 | 3E–3 | 0.40 | 5.62% | 1E–3 | 0.31 | 149258 | 17–20 | 1 | 0 | 1 | ↓ |
| Fusicatenibacter | 350 | 192 | 3E–3 | 0.28 | 2.15% | 3E–4 | 0.18 | 24649 | 17, 19 | 1 | 0 | 1 | ↓ |
| Harryflintia | 143 | 47 | 0.04 | 1.50 | <0.01% | 0.13 | 1.62 | 5 | | 1 | 1 | 0 | ↑ |
| Hungatella | 264 | 98 | 3E–3 | 2.31 | 0.04% | 0.02 | 2.76 | 32 | 18–20 | 1 | 1 | 0 | ↑ |
| Klebsiella | 165 | 56 | 0.01 | 2.09 | 0.78% | 0.03 | 2.85 | 13 | | 3 | 2 | 0 | ↑ |
| Lachnospiraceae unclass. | 392 | 206 | 0.01 | 0.37 | 4.05% | 1E–3 | 0.24 | 28538 | 17–20 | 1 | 0 | 1 | ↓ |
| Lactobacillus | 283 | 95 | 5E–6 | 4.55 | 0.27% | 1E–5 | 7.16 | 44 | 17–19 | 10 | 6 | 0 | ↑ |
| Megasphaera | 116 | 25 | 1E–3 | 2.34 | 0.12% | 1E–3 | 3.37 | 3 | 18, 19 | 2 | 2 | 0 | ↑ |
| Methanobrevibacter | 185 | 63 | 5E–3 | 2.65 | 0.25% | 0.01 | 3.16 | 19 | 19 | 1 | 1 | 0 | ↑ |
| Methanomassiliicoccus | 45 | 9 | 0.02 | 1.34 | 0.01% | 0.08 | 1.51 | 2 | | 1 | 1 | 0 | ↑ |
| Monoglobus | 178 | 109 | 0.08 | 0.54 | 0.18% | 0.01 | 0.30 | 124 | 19 | 1 | 0 | 1 | ↓ |
| Porphyromonas | 66 | 10 | 2E–3 | 1.97 | <0.01% | 5E–3 | 1.80 | 1 | 17, 19 | 1 | 1 | 0 | ↑ |
| Roseburia | 437 | 223 | 1E–4 | 0.31 | 3.71% | 4E–7 | 0.21 | 168192 | 17–20 | 6 | 0 | 3 | ↓ |
| Ruminococcus | 338 | 180 | 0.03 | 0.35 | 3.65% | 7E–3 | 0.24 | 16004 | | 4 | 0 | 3 | ↓ |
| Ruthenibacterium | 479 | 228 | 5E–6 | 2.17 | 0.50% | 2E–3 | 1.80 | 28379 | | 1 | 1 | 0 | ↑ |
| Scardovia | 50 | 5 | 2E–3 | 1.47 | <0.01% | 7E–3 | 1.56 | 1 | | 1 | 1 | 0 | ↑ |
| Turicibacter | 105 | 24 | 7E–4 | 2.14 | 0.01% | 1E–3 | 2.46 | 2.3 | | 1 | 1 | 0 | ↑ |
| **(vii) Association at species-level, missed at genus-level** | | | | | | | | | | | | | |
| Alistipes | 461 | 215 | 0.23 | 1.45 | 4.77% | 0.62 | 1.25 | 140168 | 18, 19 | 7 | 2 | 0 | Missed |
| Bacteroides | 489 | 234 | 0.86 | 1.02 | 29.33% | 0.25 | 0.83 | 3556422 | 18 | 21 | 1 | 0 | Missed |
| Dialister | 154 | 85 | 0.33 | 0.73 | 0.12% | 0.17 | 0.50 | 37 | 18 | 2 | 0 | 1 | Missed |
| Lachnoclostridium | 460 | 211 | 0.08 | 1.52 | 0.61% | 0.33 | 1.45 | 12245 | | 12 | 2 | 0 | Missed |
| Lachnospira | 139 | 78 | 0.36 | 0.73 | 0.28% | 0.18 | 0.49 | 42 | 17 | 2 | 0 | 1 | Missed |
| Parabacteroides | 464 | 218 | 0.16 | 1.45 | 3.10% | 0.58 | 1.25 | 133559 | 18, 20 | 4 | 1 | 0 | Missed |
| Prevotella | 248 | 131 | 0.19 | 0.58 | 4.24% | 0.12 | 0.41 | 288 | 18 | 2 | 0 | 1 | Missed |
| Pseudoflavonifractor | 93 | 35 | 0.16 | 1.33 | 0.01% | 0.50 | 1.26 | 3 | | 2 | 1 | 0 | Missed |
| Rothia | 184 | 99 | 0.16 | 0.69 | 0.02% | 0.07 | 0.49 | 31 | | 2 | 0 | 1 | Missed |
| Ruminococcaceae unclass. | 460 | 211 | 0.11 | 1.60 | 2.32% | 0.30 | 1.46 | 24992 | 19, 20 | 5 | 2 | 0 | Missed |
| Streptococcus | 479 | 229 | 0.26 | 1.26 | 1.79% | 0.96 | 1.03 | 66936 | 20 | 14 | 4 | 4 | Missed |
| Veillonella | 116 | 67 | 0.36 | 0.79 | 0.06% | 0.18 | 0.55 | 14 | 18 | 4 | 0 | 1 | Missed |

MWAS was conducted to test differential abundance of genera in 490 PD and 234 NHC. 107 of 224 detected genera were tested (excluding genera found in <5% of samples). Two statistical tests were used in parallel: ANCOM-BC and MaAsLin2. Analyses were adjusted for stool sample collection method and total sequence count per sample. (i) 34 genera emerged from MWAS as PD-associated, based on reaching significance by both MaAsLin2 and ANCOM-BC (i.e., FDR < 0.05 by one method and FDR≤0.1 in the other). (ii) Prevalence of the genus, i.e., the number of subjects in whom the genus was detected (out of 490 PD and 234 neurologically healthy controls (NHC)). (iii and iv) For each PD-associated genus, we show the results from MaAsLin2 and ANCOM-BC, including FDR for statistical significance of association, the fold change (FC) in abundance of the genus in PD vs. NHC, and the abundance of genus in NHC, which is relative abundance (RA) for MaAsLin2, and sampling bias-corrected observed abundance (BC-OA) for ANCOM-BC. Full genus-level MWAS results, including standard errors, are provided in Supplementary Data 2. (v) Present study, conducted using shotgun sequencing, confirmed prior findings that were based on 16S rRNA gene sequencing. Studies that have reported association of PD with the genus are cited. Because of limited sample sizes of individual studies, here, we refer to three meta-analyses and our most recent study, which was not included in the meta-analyses (each with *N* > 500). (vi) Breakdown of each genus to species resolution to evaluate heterogeneity. For each PD-associated genus, we show the number of species that were included in species-level MWAS, and how many were found significantly elevated (up arrow) or reduced (down arrow) in PD, and what the cumulative effect was, seen as presence/absence and direction of the signal detected at genus level. An example of heterogeneity is *Actinomyces*. Ten *Actinomyces* species were tested in species-level MWAS, 5 were found to be associated with PD, 4 were elevated and 1 was reduced. At genus-level MWAS, *Actinomyces* was elevated in PD. Similarly, *Eubacterium*, which at genus-level MWAS was found to be decreased in PD, had two species that were elevated in PD. (vii) Genera that did not show a signal for association with PD at genus-level MWAS, but some of their species were associated with PD per species-level MWAS. Most notably, the *Streptococcus* genus, with 8 PD-associated species (4 elevated, 4 reduced in PD), was missed at genus level.

overlap (i.e., FDR < 0.05 by both methods) was 15% of KO groups and 32% of pathways. We therefore estimate that between 1/3 to 2/3 of all detected metabolic pathways are dysregulated in PD.

## Functional inference

Much of the constituents of the gut microbiome have been detected by metagenomics and not yet been characterized, nor can we infer on KO groups that have not yet been annotated. Thus, inferences made here are robust given the current state of knowledge and available resources, but do not capture the full potential of the study. While many of the pathways and KO groups that were altered in PD are non-specific and likely reflect broad dysbiosis, we noted several features in the PD metagenome that align with pathological features of PD (Fig. 5). Most functional insights to PD pathogenesis are derived from model organisms and in experimental setting, with the caveat that their relevance to human PD is not guaranteed. Here, we provide data from the human PD gut metagenome that corroborate, and validate, some of the basic science findings.

As detailed below, we found that PD metagenome is indicative of a disease promoting microbiome (enriched in opportunistic pathogens and immunogenic components, dysregulated neuroactive signaling, preponderance of amyloidogenic molecules that induce alpha-synuclein pathology, and over production of toxicants), with reduced capacity for recovery (low in anti-inflammatory and neuro-protective molecules).

We observed a general increase in PD metagenome of pro-inflammatory species, as well as gene-families and pathways that directly promote inflammatory signaling between microbes and the host. Inflammation has emerged as a major driver of PD pathogenesis[33]. Lipopolysaccharides (LPS) are the most abundant antigen on the surface of Gram-negative bacteria, capable of eliciting strong immune reaction and inflammation, by stimulating the host's Toll-like Receptor 4 (TLR4) signaling pathway. We detected a bloom of Gram-negative bacteria with immune-stimulatory LPS in PD metagenome. Eleven of the 55 species that are enriched in PD are canonical Gram-negative organisms including those with highly stimulatory LPS (*Escherichia coli*, *Klebsiella* species, and *Porphyromonas asaccharolytica*). Conversely, of the 29 species that are reduced in PD, only 1 is a canonical Gram-negative organism, *Prevotella copri*, which interestingly, produces an LPS that not only does not induce inflammation, but can inhibit TLR4 activation by others[34]. We also detected elevated levels of gene families and pathways necessary for generation and secretion of Lipid A, a glycolipid component of LPS responsible for much of its immune-stimulatory capacity. The PD metagenome was enriched for the KO groups in the canonical (Raetz Pathway) lipid A synthesis pathway including *lpxC* (K02535: FC = 2.3, 95% CI = 1.5–3.5, FDR = 1E−3), *lpxI* (K09949: FC = 1.7, 95% CI = 1.3–2.3, FDR = 1E−3), *lptD* (K04744: FC = 1.9, 95% CI = 1.4–2.5, FDR = 1E−3), and the lipid A synthesis pathway as a whole (KDO-NAGLIPASYN-PWY: FC = 1.5, 95% CI = 1.1–2, FDR = 0.02). We further noted an enrichment of the synthesis pathway for the core, immunogenic sugar components of LPS (ECASYN-PWY: FC = 1.7, 95% CI = 1.2–2.5, FDR = 5E−3), and the general fatty acid synthesis pathways necessary for production of the Gram-negative cell envelope (PWY-6285 and PWY0-881: FC = 1.6, 95% CI = 1.2–2, FDR = 1E−3). We observed increases in gene-families involved in the synthesis of Lipoteichoic acid (LTA), a component of Gram-positive bacteria cell envelope which shares many of the pathogenic properties of LPS (K19005: FC = 1.8, 95% CI = 1.4–2.3, FDR = 4E−5), and in its modification to be recognized by host TLR2 and stimulate inflammatory responses[35] (K03739: FC = 1.7, 95% CI = 1.3–2.2, FDR = 1E−3). Additionally, we observed enrichment in PD of murein/bacterial lipoprotein (BLP) gene-family [*lpp*] K06078 (FC = 1.6, 95% CI = 1.2–2.1, FDR = 0.01). BLP is highly immune stimulatory by triggering inflammatory TLR2 signaling[36], but is often overlooked comparatively to LPS.

We detected reduction in species, genes and pathways that degrade polysaccharides and produce SCFA. SCFAs are products of fermentation of dietary fiber by the gut microbiome. In the gut, inadequate SCFA levels have been linked to constipation, compromised gut barrier, and inflammation[37,38], all of which are common features of PD. This dataset revealed depletion of species, genes and pathways that metabolize plant-based polysaccharides and make SCFA. We observed a dramatic ~2.5-fold reduction for several SCFA producing species, and as much as 5 to 7.5 fold reduction in *Roseburia* species. At gene-family level, many of the most decreased KO groups in the PD metagenome are those involved in polysaccharide utilization, each showing more than 2-fold reduction, e.g., *araQ* (K17236: FC = 0.45, 95% CI = 0.34–0.6, FDR = 7E-6), *araN* (K17234: FC = 0.48, 95% CI = 0.36–0.62, FDR = 9E-6), *cbe/mbe* (K16213: FC = 0.47, 95% CI = 0.34–0.67, FDR = 4E-4), and E2.4.1.20 (K00702: FC = 0.48, 95% CI = 0.35–0.66, FDR = 1E-4). We found a less dramatic, but statistically significant reduction in pathways critical for degradation and metabolism of plant-based polysaccharides including beta-mannan (PWY-7456: FC = 0.82, 95% CI = 0.75–0.89, FDR = 1E-4), stachyose (PWY-6527: FC = 0.83, 95% CI = 0.79–0.89, FDR = 3E-7), inositol (PWY-7237: FC = 0.81, 95% CI = 0.75–0.87, FDR = 7E-7), fructuronate (PWY-7242: FC = 0.83, 95% CI = 0.76–0.9, FDR = 1E-4), galacturonate (GALACTUROCAT-PWY: FC = 0.89, 95% CI = 0.82–0.95, FDR = 3E-3), glucuronate (GLUCUROCAT-PWY: FC = 0.84, 95% CI = 0.78–0.91, FDR = 1E−4), and galacturonate/glucoronate (GALACT-GLUCUROCAT-PWY: FC = 0.85, 95% CI = 0.79–0.92, FDR = 5E-4). Interestingly, the PD metagenome was enriched in glycogen metabolism pathway (GLYCOCAT-PWY: FC = 2, 95% CI = 1.5–2.7, FDR = 7E-5), suggesting an enrichment in bacteria with a preference for non-plant-based polysaccharides. Of note, unlike most polysaccharide degradation pathways that are reduced, the starch degradation pathway III is enriched in the PD metagenome (PWY-6731: FC = 2, 95% CI = 1.5-2.6, FDR = 7E-5), however, pathway III maps to *Escherichia coli* and *Klebsiella* species and is not known to produce beneficial SCFA.

Proteolytic and amino acid degradation pathways were substantially enriched in PD metagenome, suggesting PD microbiome preferentially utilizes proteins, over polysaccharides, as their prime carbon/energy source. All proteolytic pathways that were differentially abundant in PD vs. NHC were enriched in the PD metagenome, including pathways which degrade glutamate (PWY-5088: FC = 1.3, 95% CI = 1.1–1.6, FDR = 3E-3), arginine/ornithine (ORNARGDEG-PWY: FC = 1.9, 95% CI = 1.3–2.6, FDR = 2E-3; ARGDEG-PWY: FC = 1.9, 95% CI = 1.3–2.6, FDR = 8E-3; ORNDEG-PWY: FC = 1.7, 95% CI = 1.3–2.3, FDR = 2E-3), and threonine (THREOCAT-PWY: FC = 1.9, 95% CI = 1.4–2.7, FDR = 5E-4). These bacterial pathways, and specifically the upregulated threonine pathway, utilize protein sources derived from diet or the host, including mucin. Mucin is an important component of the protective gut mucus layer. Increased capacity to degrade host mucin raises the possibility that the gut microbiome contributes to the degradation and increased permeability of the gut barrier in PD[39].

We detected dysregulation in synthesis and metabolism of neuroactive molecules dopamine, glutamate, gamma aminobutyric acid (GABA), and serotonin. Aberrations involving these neurotransmitters in PD is well established[40]. The human gut microbiome has known capacity to not only produce and metabolize these neurotransmitters in the gut, but also to influence their levels in the brain[41]. A substantial drop in dopamine levels due to progressive loss of dopaminergic cells in the brain is the defining feature of PD. We detected elevated levels of tyrosine decarboxylase/aspartate 1-decarboxylase gene-family (K18933: FC = 1.57, 95% CI = 1.3–1.9, FDR = 3E−4), which removes tyrosine, a required precursor of dopamine. Thus, one may infer that elevated K18933 could limit dopamine production by host and bacteria. The aromatic amino acid synthesis pathway (COMPLETE-ARO-PWY: MaAsLin2 FC = 0.95, 95% CI = 0.92–0.98, FDR = 6E−3; ANCOM-BC FC = 0.74, 95% CI = 0.67–0.82, FDR = 1E-8) was depleted in PD

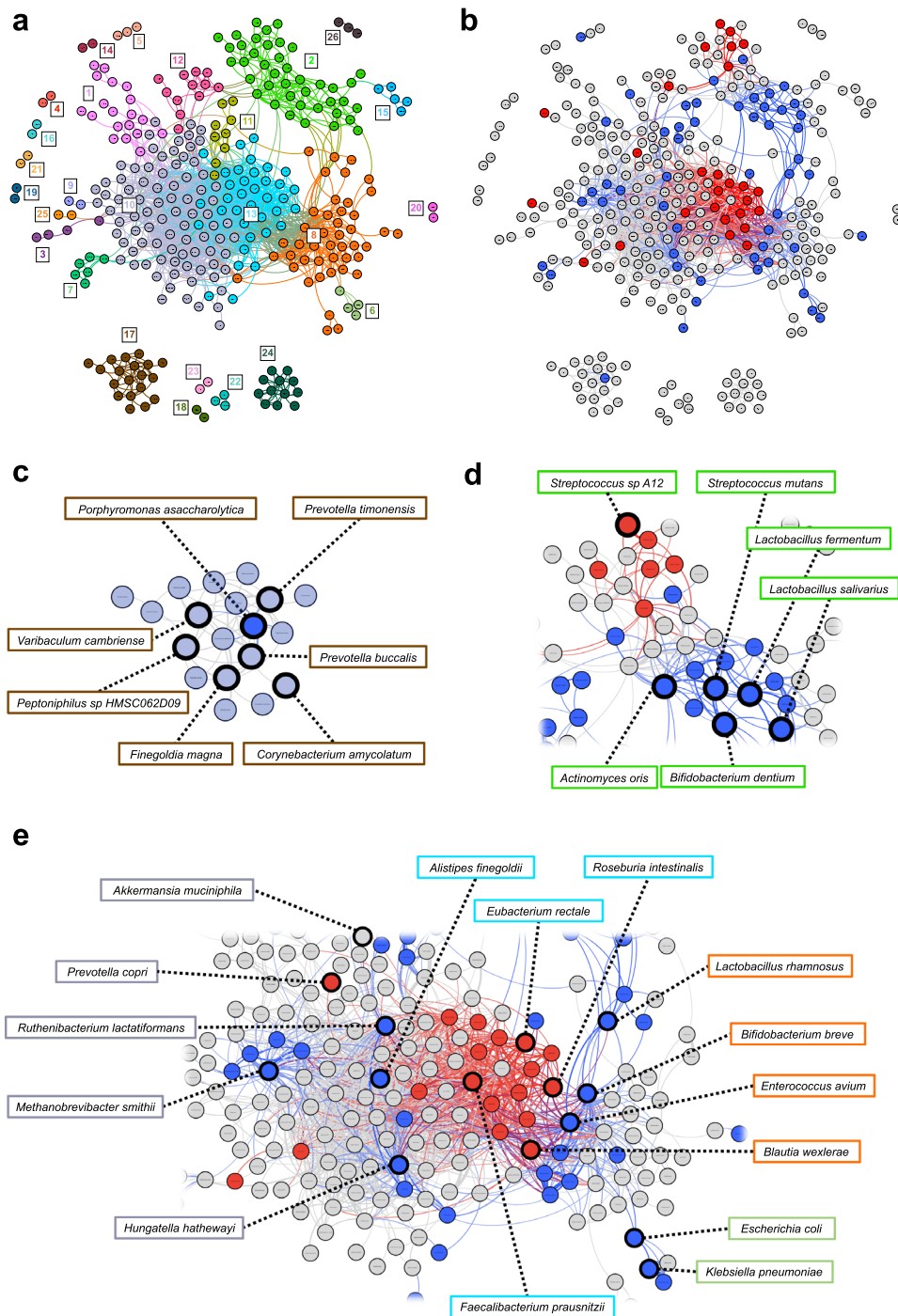

**Fig. 4 | Network analysis reveals polymicrobial clusters of correlated species in the PD metagenome.** Analysis included $N = 490$ biologically independent samples from 490 PD cases. (For neurologically healthy control (NHC) network, $N = 234$, see Supplementary Fig. 4). **a** All species detected in PD gut metagenome were tested for correlation with one another using SparCC correlations and plotted in a network if their abundance correlated with at least one other species (i.e., $|r| > 0.2$ and uncorrected permuted $P$-value <0.05). Clusters were defined by Louvain algorithm and were randomly assigned a color and a number. Each circle (node) denotes a species and the curved lines (edges) connect correlated species. **b** PD-associated species that were identified via MWAS were mapped to the network, and highlighted in blue if elevated in PD, or red if reduced in PD. Correlation among PD-associated species were often positive, indicating abundances tended to rise together like *Escherichia coli*, *Klebsiella pneumoniae*, and *Klebsiella quasipneumoniae* in cluster #6, or decline together like the polysaccharide metabolizing species in cluster #13. Sometimes the correlation was negative indicating increase in abundance of one species correlated with decrease in the other, e.g., in cluster #2 the rise in abundances of some *Streptococcus* species in PD microbiome correlated with a decline in the abundances of other *Streptococcus* species. Enlargements with representative species labeled for (**c**) cluster #17, (**d**) cluster #2, and (**e**) cluster #8 labeled to the left, cluster #13 in the middle, cluster #8 on the upper left, and cluster #6 on the lower left (*E. coli* and *Klebsiella* species). For (**c**–**e**), boxes around species names are colored to represent their algorithm defined cluster shown in (**a**).

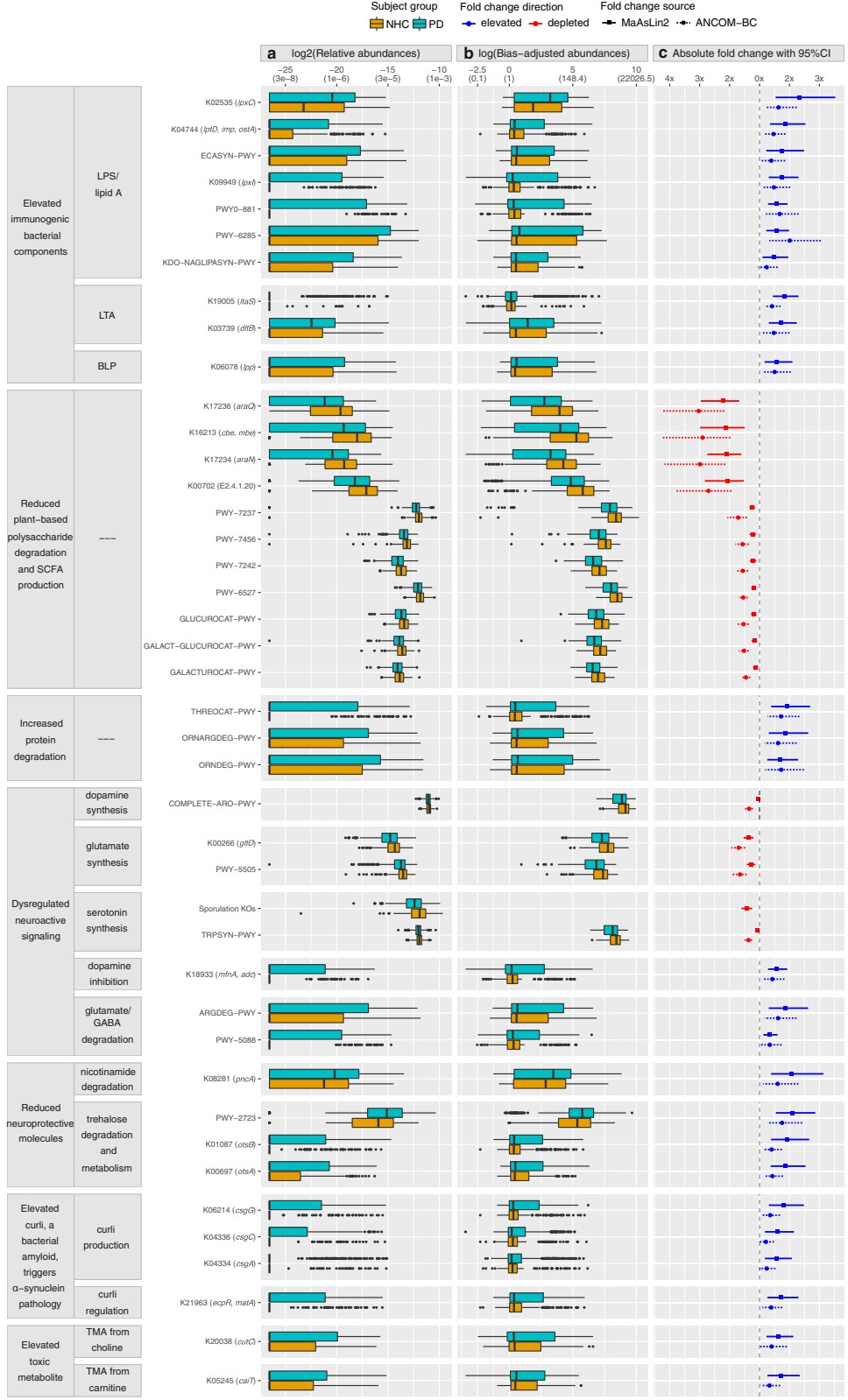

metagenome, suggesting that the microbiome has diminished capacity to generate precursors to dopamine and possibly other catecholamines such as adrenaline.

Tyrosine decarboxylase (TDC) is noteworthy because it is a bacterial enzyme capable of metabolizing L-dopa to dopamine in the gut before it can reach the brain. TDC is encoded by bacterial *tdc* gene. *Enterococcus faecium* and *Enterococcus faecalis tdc*

genes metabolize L-dopa efficiently, and their abundance in the gut correlates directly with reduced efficacy of L-dopa for the patient[42,43]. We did not detect the *tdc* gene or the TDC pathway in our dataset, not necessarily because they do not exist but more likely that they were filtered out by the stringent QC criteria. We did however detect the species, *E. faecium* and *E. faecalis*, and both were elevated in PD. *E. faecium* rose to significance in MWAS

**Fig. 5 | Microbial gene-families and pathways with functional relevance to PD.**
Analysis included *N* = 724 biologically independent samples from 490 PD and 234 neurologically healthy control (NHC) subjects. Overall, 15% of microbial gene-families (KO groups) and 30% of pathways (MetaCyc) tested were elevated or depleted in PD, a conservative estimate derived from consensus at FDR < 0.05 by two statistical methods (MaAsLin2 and ANCOM-BC) (Supplementary Data 9, 10). Examples are shown here, grouped by inferred functional relevance to PD (left panel). Data show increased levels of microbial activities that could contribute to PD pathogenesis (immunogenicity, alpha-synuclein aggregation, and creation of toxic metabolites), and reduced levels of protective mechanisms (anti-inflammation, and neuroactive and neuroprotective molecules). **a** Relative abundances (used in MaAsLin2). **b** Bias-corrected observed abundances (used in ANCOM-BC). **c** Fold change in PD vs. NHC as estimated by MaAsLin2 and ANCOM-BC. *Y*-axis: KO groups (identifiers begin with "K", gene symbol in parentheses) and pathways ("PWY"). *X*-axis: Log2 transformed relative abundances (used in MaAsLin2) (**a**), or natural log transformed bias-corrected observed abundances (used in ANCOM-BC) (**b**), with

untransformed relative abundances in parenthesis for easier interpretation; (**c**) fold change in differential abundance in PD vs. NHC (squares and circles) with 95% confidence interval (CI; solid and dashed lines), calculated from beta and standard errors estimated by MaAsLin2 and ANCOM-BC. Content in (**a** and **b**): Boxplots show frequency distribution of each KO and pathway in PD (blue green) and NHC (orange) metagenomes. Left, middle, and right vertical boundaries of each box represent first, second (median), and third quartiles of the data. The absence of a box indicates >75% of samples had zero abundance. The lines extending from the two ends of each box represent 1.5x outside the interquartile range (range=(abundance value at 75% minus abundance value at 25%)x1.5). Points beyond the lines are outlier samples. Content in (**c**): Fold change difference between PD and NHC in log2-transformed relative abundance (square with solid line of 95% CI, MaAsLin2), or natural log transformed bias-corrected observed abundances (circle with dotted line of 95% CI, ANCOM-BC). Blue: elevated in PD, red: reduced in PD. LPS: Lipopolysaccharide. LTA: Lipoteichoic acid. BLP: murein/bacterial lipoprotein. SCFA: short-chain fatty-acid. TMA: Trimethylamine.

and was nominated as PD-associated species with FC = 1.4 (95% CI = 1.1–1.8) by MaAsLin2 and FC = 1.8 (95% CI = 1.2–2.8) by ANCOM-BC, both with FDR-0.02. *E. faecalis* yielded FC = 1.2 (95% CI = 1–1.5) with FDR = 0.1 by MaAsLin2 and FC = 1.3 (95% CI = 0.94–1.9) with FDR = 0.2 by ANCOM-BC.

The link between PD and glutamate toxicity due to overactivation of glutamate receptor has been demonstrated experimentally[44] and associated to glutamate receptor gene in humans[45]. Imbalance between GABA and glutamate (excitation/inhibition imbalance) weakens synaptic connection and promotes neurodegeneration[46]. In the PD metagenome, we noted a decrease in glutamate/glutamine synthesis gene-families and pathways (K00266: FC = 0.74, 95% CI = 0.65–0.83, FDR = 5E−5; PWY-5505: FC = 0.78, 95% CI = 0.7–0.87, FDR = 1E−4), and a concomitant increase in both glutamate and GABA degradation (PWY-5088: FC = 1.3, 95% CI = 1.1–1.6, FDR = 2E−3; ARG-DEG-PWY: FC = 1.9, 95% CI = 1.3–2.6, FDR = 2E−3).

The highest concentration of serotonin is in the gut. Serotonin abnormalities have been detected at various stages of PD, from before the development of motor symptoms, making it a potential early diagnostic marker[47], to late stages of PD where it may be a risk factor for cognitive decline[48]. It was shown in mice that spore forming bacteria induce production of serotonin in host colonic cells[49]. Tryptophan, an essential amino acid only derived from microbiome or diet, is the rate limiting precursor to serotonin. Tryptophan biosynthesis pathway was reduced in PD (TRPSYN-PWY: MaAsLin2 FC = 0.94, 95% CI = 0.9–0.98, FDR = 9E−3; ANCOM-BC FC = 0.73, 95% CI = 0.67–0.8, FDR = 7E−10). We also detected a near global reduction in sporulation genes in PD. We tested 33 sporulation KO groups in this dataset, 31 of them were depleted in PD (0.49 ≤ FC ≤ 0.77, 2E−5 ≤ FDR ≤ 0.04), suggesting that intestinal serotonin production in the host may be diminished.

Serotonin affects gut motility through modulation of intestinal contractility[50]. Given that spore forming bacteria modulate serotonin production and enhance gut motility[49,50], and our data suggest a global depletion of the sporulation KO groups, we speculated, and tested the hypothesis that constipation, a common symptom of PD, may be related to the depletion of spore forming bacteria. We selected 27 sporulation KO groups that were FDR < 0.05 in both ANCOM-BC and MaAsLin2 and tested the hypothesis that the relative abundance of sporulation KO groups is even lower in subjects who reported constipation vs. those that did not. Indeed, lower abundances of sporulation KO groups was associated with constipation both in PD group (FC = 0.76, 95% CI = 0.66–0.89, *P* = 2E−4), and in NHC group (FC = 0.58, 95% CI = 0.41–0.83, FDR = 3E−3). We should note that association of sporulation KO groups with PD is not solely due to constipation; adjusting for constipation, the relative abundance of sporulation KO groups was still significantly lower in PD than in NHC (FC = 0.79, 95% CI = 0.68-0.9, *P* = 5E−4).

Nicotinamide and trehalose are neuroprotective molecules. In PD, disruption of energy homeostasis leads to dysregulation of nicotinamide production and promotes neurodegeneration[51]. A recent clinical trial of PD with nicotinamide supplementation reported a significant improvement in pathological indicators including inflammation[52]. We observed a two-fold increase in the nicotinamidase gene family (K08281: FC = 2.1, 95% CI = 1.4–3.1, FDR = 3E−3), suggesting the PD gut microbiome degrades this neuroprotective molecule.

The disaccharide trehalose has garnered significant interest as a therapeutic molecule in PD and other neurodegenerative diseases[53]. This is likely due to its activities on autophagocytic pathways which limit the accumulation of pathogenic alpha-synuclein and other protein aggregates[54]. As we note with nicotinamide degradation, a similar two-fold increase was observed in the PD metagenome of the trehalose degradation pathway (PWY-2723: FC = 2.1, 95% CI = 1.5-2.9, FDR = 4E-5). We also noted evidence for increased metabolism of trehalose by bacteria (K00697: FC = 1.9, 95% CI = 1.4-2.5, FDR = 1E-3 and K01087: FC = 1.9, 95% CI = 1.4-2.7, FDR = 1E-3), a potential outcome of which is reduced availability of trehalose for the host.

The bacterial-derived amyloidogenic protein, curli, exacerbates alpha-synuclein pathology and inflammation in experimental models of PD[14–16]. Many *Enterobacteriaceae* species encode curli[55]. We detected 33 species in *Enterobacteriaceae* family but only *Escherichia coli* and *Klebsiella* species had high-enough abundance to be tested, and they were all elevated in PD. We also found an enrichment of the gene-families related to curli. *csgA* (K04334: FC = 1.6, 95% CI = 1.2–2.1, FDR = 9E−3), *csgB* (K04335: FC = 1.3, 95% CI = 1–1.5, FDR = 0.04), and *csgC* (K04336: FC = 1.6, 95% CI = 1.2–2.2, FDR = 0.01) are central to curli structure. *csgG* (K06214: FC = 1.8, 95% CI = 1.3–2.5, FDR = 3E−3) is necessary for the extracellular secretion and synthesis of curli. LuxR/Mat/Ecp (K21963: FC = 1.7, 95% CI = 1.3–2.3, FDR = 3E−3) is a family of transcriptional factors that regulate the expression of curli.

Trimethylamine (TMA) is a toxic metabolite produced by gut microbiome from foods such as red meat and egg yolk. High level of circulating TMA has been linked to increased risk of cardiovascular disease and stroke, likely due to toxic and inflammatory effects[56]. Circulating levels of TMAO, a gut microbiome derived oxidation product of TMA, are reportedly elevated in PD and correlate with disease progression and severity[57]. Gene families involved in TMA production were elevated in PD, including *cutC* (choline lyase) which cleaves choline to produce TMA (K20038: FC = 1.62, 95% CI = 1.2–2.1, FDR = 4E−3), and *caiT* (L-carnitine/gamma-butyrobetaine antiporter) which exchanges carnitine for gamma-butyrobetaine before TMA can be produced (K05245: FC = 1.7, 95% CI = 1.3–2.3, FDR = 4E−3).

In summary, we have generated, and share publicly, a large dataset at the highest resolution currently feasible. The dataset itself is a substantial contribution to open science because it includes individual-level deep shotgun metagenome sequences and extensive

metadata on 490 persons with PD, the largest PD cohort with microbiome data, and a unique cohort of 234 neurologically healthy elderly, which can be used in a wide range of studies. We uncovered a widespread dysbiosis in PD metagenome that is indicative of an environment permissive for neurodegenerative events and prohibitive of recovery. This study exemplified bi-directional transfer of information between human and experimental studies: The data and results generated here from human metagenomes sets a broad foundation for basic research, including numerous concrete hypotheses which will be tested experimentally to discern the multitudes of roles that the gut microbiome may play in PD. At the same time, the results generated here confirmed, in humans, several observations that were made experimentally in animal models. While this study generated many hypotheses with solid evidence that can be tested now, it did not achieve the full potential of a metagenomics study. Metagenomics is a new, albeit fast evolving field, and the resources, methods, and tools, while state of the art, are still in development. Undoubtedly more information will be revealed as we increase the sample size, and others also conduct metagenomics studies and share the data. We anticipate that in near future we will have the tools and the analytic power to use metagenomics, as a new approach, to study PD heterogeneity, search for biomarkers, delve deeper into the origin and progression of PD sub-phenotypes, and investigate the potential in manipulating the microbiome to prevent, treat and halt the progression of PD.

## Methods

We have complied with all relevant ethical regulations. Study was approved by the Institutional Review Board (IRB) for Protection of Human subjects at the University of Alabama at Birmingham (UAB) and by the Human Research Protection Office (HRPO) of United States Department of Defense (funding agency). All subjects gave signed informed consent, approved by UAB IRB and DoD HRPO. No compensation was provided for participating in the study.

Study was conducted at the University of Alabama at Birmingham (UAB) by a single team of NGRC investigators from start (study design and subject enrollment) to finish (analysis and interpretation), ensuring uniform methods. The same investigator team conducted the two prior studies that are being referenced here for comparison of shotgun and 16S analyses.

Software and reference databases are referred to by name only in the text; the versions and the URLs are provided in Supplementary Data 11. The workflow is in Fig. 1. Metadata collection, DNA extraction, sequencing and QC, taxonomic and functional profiling, and statistical analyses (MWAS) were blinded.

### Study cohort

**Enrollment.** 670 individuals with PD and 316 NHC were enrolled at UAB between October 2018 and March 2020. All subjects were from the same geographic region, including the city of Birmingham and surrounding areas in the southern US, keeping confounding effect of geography on the microbiome to a minimum. The eligibility criteria to enroll as a case was diagnosis of PD and informed consent. Potential PD cases for enrollment were identified via systematic pre-screening of electronic medical records (EMR) of patients with an upcoming appointment in the Movement Disorder Clinic at UAB. Subjects were invited to enroll in the study after their clinic visit if the attending specialist confirmed PD diagnosis and the patient was willing to hear about the study. With each potential participant, in a private setting, our recruitment coordinator explained the study and their involvement, subject was given time to read the Informed Consent Form, to ask questions, and if they agreed to participate, sign the consent. During enrollment visit, subjects donated a blood or saliva sample, and took home two questionnaires and a stool collection kit to complete and return in the pre-stamped envelope via US Postal Service. The spouse or friend accompanying the patient was invited to enroll as

control, to match for shared environmental effects. Additional control subjects were recruited from the community. Consent, enrollment, and data collection process for controls was the same as for patients if the control was met in person. For community volunteers, we first had a phone conversation to explain the study, then we mailed the Informed Consent Form to them with a phone number to call with any questions. Those who returned a signed Informed Consent Form were sent a kit for saliva collection, a kit for stool collection, and two questionnaires to complete at home and return by mail. The eligibility criteria to enroll as a control was absence of PD, REM sleep behavior disorder, Alzheimer's disease, dementia, multiple sclerosis, amyotrophic lateral sclerosis, ataxia, dystonia, autism, epilepsy, stroke, bipolar disorder, and schizophrenia by self-report (hence, neurologically healthy controls), preferably being over age 50 years, and informed consent. We tried not to enroll subjects who had participated in our prior microbiome studies to maintain cohort independence; however, 11 subjects were inadvertently double-enrolled. We ended the enrollment at the start of the COVID-19 pandemic in the southern US (March 2020) to avoid confounding by infection or stress of the pandemic.

**Exclusions.** Subjects who did not return a stool sample were excluded (171 PD, 73 controls). One PD subject was excluded because the stool sample was received during the pandemic. We excluded 1 control for potential sample mix-up and 8 for having neurologic disease (7 stroke, 1 epilepsy). Since the diagnosis of PD can change with time, we reviewed EMR prior to sequencing, and again before data analysis, and those whose diagnosis changed were excluded ($N = 6$). One PD subject was excluded for low sequence count. The final sample size was 490 PD and 234 NHC.

### Metadata and stool sample collection

Subjects were given a packet containing two questionnaires (metadata), and a stool collection kit with detailed instructions and a self-addressed pre-stamped envelope for return.

**Metadata.** The Environmental and Family History Questionnaire (EFQ) was used to collect PD-related data, and the Gut Microbiome Questionnaire (GMQ)[17,28] was filled out by subject immediately after stool sample collection and gathered gut-related data. The questionnaires have been provided in Supplementary Fig. 1. Questionnaires were completed by subject without investigator interference.

**Stool sample.** Each subject provided one stool sample at one time point; hence each data point represents a unique individual. Each sample was measured and analyzed once. Stool samples were collected at home. Nearly all subjects (487 PD, 232 NHC) used the DNA Genotek (Ottawa, Ontario, Canada) OMNIgene GUT Collection kit. Five subjects (3 PD, 2 NHC) used Fisher Scientific (Hampton, NH, USA) DNA/RNA-free BD BBL Sterile/Media-free Swabs. Collection method was included as a technical covariate in analyses. Upon receipt in the lab, samples were stored in −20 °C freezer.

**QC of returned packets.** Upon receipt of each packet, the EFQ, GMQ and stool sample in the packet were cross-checked for potential labeling error in the lab or sample mix-up at home (PD and control pairs living together) and for completeness, comprehensibility, and lucidity of information. Issues were resolved by checking the EMR and calling the subject, and if not resolved, the sample or metadata in question were excluded. Metadata were entered in a customized PROGENY database software (Progeny Genetics LLC, Aliso Viejo, CA, USA), and in Excel spreadsheets, by two data entry staff, downloaded, cross-checked, and errors corrected. PROGENY was used for storage and retrieval of data.

**DNA isolation, library preparation, next-generation sequencing**
Prior to shipping samples for processing, case and control samples were inter-mixed to avoid batch effect during DNA isolation and sequencing. Frozen stool samples were shipped on dry ice to the metagenomic services company CosmosID (Germantown, MD, USA). DNA was isolated from an aliquot of stool sample using the Qiagen (Germantown, MD, USA) DNeasy PowerSoil Pro high-throughput kit on the QIAcube high-throughput platform according to the manufacturer's protocol. Isolated DNA was quantified by Qubit. The sequencing libraries were prepared by using the Illumina Nextera XT transposase system. Libraries were sequenced on the Illumina NovaSeq 6000 instrument aiming for 40 M reads per sample, following paired-end 150 bp sequencing protocol. Positive and negative controls and technical replicates were included for extraction and sequencing, all passed QC. The number of raw sequence reads per sample ranged within 17 M to 385 M (average 50 M per sample).

**Bioinformatic processing of sequences**
**QC of sequences**. QC of sequence reads was performed using BBDuk and BBSplit. In the first step of QC, BBDuk was used to remove Nextera XT adapter and PhiX genome contamination, and quality trim and filter sequences. BBDuk was ran with the 'ftm=5' to ensure any extra bases after the 150 bp were removed, 'tbo' option to additionally trim adapters based on paired read overlap detection, 'tpe' option to trim paired reads to the same length, 'qtrim=rl' to quality trim both 5′ and 3′ ends of sequences, 'trimq=25' to trim ends of sequences up to quality score of 25, and 'minlen=50' to remove sequences that fall below 50 bp after contaminate removal and trimming. Percent of sequences removed ranged from 22% to 72% of initial reads with a mean of 28% removed per sample. Next, all sequence reads mapping to human reference genome (GRCh38.p13) were removed using BBSplit with default parameters, eliminating <3% of BBDuk processed reads for most samples. Finally, we removed low-complexity sequences (<2% of human decontaminated reads) with BBDuk entropy filtering specifying 'entropy=0.01', which was performed to remove sequencing artifacts consisting of overrepresented long N-mers of "G". The resulting QC'ed sequences consisted of 3 M to 258 M reads (average 36 M) per sample. One sample with 3 M reads was removed for analysis, but is included in the publicly available dataset. In essence, we followed the same QC protocol as recommended by HMP[24], with one difference: as our sequencing protocol did not include PCR amplification step, we did not remove duplicated sequences, as these may be actual biological overrepresentation.

**Taxonomic profiling**. QC'ed sequences were profiled using MetaPhlAn3[25] with the accompanying database of ~1.1 M unique clade-specific marker genes. MetaPhlAn3 was ran twice: (1) relative abundances were generated using the default settings to be used in downstream analyses with MaAsLin2[58], and (2) relative abundances were generated using the '--unknown-estimation' flag to be multiplied by the total sequence reads of the sample to generate counts used in ANCOM-BC[59]. Taxonomic profiling resulted in 2,270 species represented by at least one marker gene, which were then trimmed by default parameters of MetaPhlAn down to 719 species (default parameters: t=rel_ab, perc_non-zero=0.33, stat_q = 0.2, avoid_disqm=FALSE, stat=tavg_g, min_cu_len=2000, unknown_estimation=FALSE).

**Enterotype profiling**. Using relative abundances of genera, the web-based EMBL enterotype classifier was used to assign enterotypes to each sample based on the original enterotype definition (*Bacteroides, Firmicutes, Prevotella*)[60]. The frequency of each enterotype in PD and NHC were calculated separately, as number of samples assigned to each enterotype divided by total number of samples.

**Functional profiling**. QC'ed sequences were profiled for potential functional content (UniRef90 gene-families[61] and MetaCyc metabolic pathways[62]) using HUMAnN3[25] with default parameters. Resulting UniRef90 gene-families were converted into KO groups using the HUMAnN utility tool 'humann_regroup_table'. Pathways and KO groups were filtered to contain only community-level data. Functional profiling resulted in 8,528 distinct KO groups and 511 metabolic pathways, numbers which are in line with previous human gut microbiome studies[25].

**Statistical analysis**
The sample size for all statistical analyses included 490 PD and 234 NHC, except tests involving constipation included 468 PD and 225 NHC, and confounding analysis requiring data on all confounders had 435 PD and 219 NHC. When using R, default parameters were used unless noted. Every test, except SparCC correlations, included technical variables as covariates. Technical variables were (1) total sequence count per sample (standardized using 'scale' in R), and (2) stool collection kit (5 subjects used sterile swabs, 719 used OMNIgene GUT). Other covariates were added as appropriate and will be noted. Significance determination was based on the test, whether agnostic or hypothesis-driven, and corrected for multiple testing accordingly, as noted below for each analysis.

**Analysis of metadata**. Distributions of 53 variables (Table 1) in PD vs. NHC were tested using Fisher's exact test for categorical variables with 'fisher.test' in R, and using Wilcoxon rank-sum test for quantitative variables with 'wilcox.test' in R. ORs and 95% CI were calculated using 'fisher.test' unless the 2×2 table contained a zero, then 'Prop.or' from the pairwiseCI R package specifying 'CImethod = 'Woolf'' was used. P-values were two-sided and not corrected for multiple testing because the aim was to detect any trend of a difference that could point to a potential confounder.

**Testing the overall composition of the metagenome**
**Principal component analysis (PCA)**. We used Aitchison distances as the measure of inter-sample differences in the compositions of gut metagenomes. Aitchison distances are Euclidean distances calculated from centered log-ratio (clr) transformed species count data. The clr transformation was performed using Eq. (1) in R:

$$\text{clr} = \log(x+1) - \text{mean}(\log(x+1)) \tag{1}$$

where $x$ is a vector of all species counts in a sample. PCA was performed on Aitchison distances, using the 'prcomp' in R. The first two PC's were plotted using 'autoplot' from the ggfortify R package.

**Beta-diversity**. The difference in the overall composition of PD vs. NHC metagenomes was formally tested on Aitchison distances using PERMANOVA[63] via 'adonis2' from the vegan R package. Significance of PERMANOVA was determined using 9,999 permutations, which is capped at P < 1E−4 for the highest significance.

**Dispersion**. To test homogeneity of dispersion (the assumption in PERMANOVA that groups being compared (PD vs. NHC) are similar in dispersion) we performed PERMDISP using 'betadisper' and 'permutest' functions from the vegan R package. Significance of PERMDISP was determined using 9,999 permutations, which is capped at $P < 1E−4$ for the highest significance.

**Enterotypes**. Differences in the enterotype frequencies in PD vs NHC was tested using chi-squared test. The initial test was conducted on the distribution across three enterotypes in PD vs. NHC via 'chisq.test' in R. To identify enterotypes that drive the difference, a sequential test was applied which uses chi-squared,

tags the factor contributing the largest effect to chi-squared as the primary factor, removes the primary factor, normalizes the frequencies of remaining variables (enterotypes), repeats the process until chi-squared is no longer significant[64]. Frequency distribution of enterotypes in PD and NHC were plotted using 'mosaic' from the vcd R package.

**Differential abundance analysis**
**Unbiased differential abundance analysis: MWAS.** MWAS approach was used to test differential abundances of species, genera, KO groups, and pathways in turn, in 490 PD vs. 234 NHC. Each test included all detected features that passed sequence QC and which were present in ≥5% of subjects (257 species, 107 genera, 4,679 KO groups, 407 pathways). The primary analysis was species-level MWAS. Genus-level MWAS was for comparability to past literature. KO group and pathway MWAS were for functional inference.

- *Statistical tests*. Two multivariable association tests, MaAsLin2 and ANCOM-BC, were used in parallel to ensure results were robust to methodological variations. We chose MaAsLin2 and ANCOM-BC because in comparative studies with real and simulated data, conducted by us and others[65,66], they were among the most robust in keeping false positive rate low while maintaining power[66]. MaAsLin2 is the differential abundance test in the bioBakery suite of tools, which we used for much of the bioinformatics. ANCOM-BC differs in approach, and in addition, has introduced the notion of "sampling fraction" to estimate how well the observed abundances in a sample represent the true absolute abundances in the ecosystem (human gut) and incorporates bias correction to adjust for differences in sampling fractions across samples. Thus, since MaAsLin2 tests relative abundances, and ANCOM-BC tests bias-corrected observed abundances (approximating absolute abundances), the results shown side-by-side provide different and complementary perspective on how PD and NHC metagenomes differ.
- *Multiple testing correction*. Benjamini-Hochberg FDR method[67] was used.
- *Significance*. For species-level MWAS (the primary analysis), we set statistical significance for association with PD as concordance between ANCOM-BC and MaAsLin2 with FDR < 0.05 by one and FDR≤0.1 by the other (i.e., a taxon was called significance if it achieved FDR < 0.05 by both MaAsLin2 and ANCOM-BC, or FDR < 0.05 in MaAsLin2 and FDR≤0.1 in ANCOM-BC, or FDR < 0.05 in ANCOM-BC and FDR≤0.1 in MaAsLin2). For other MWAS, while no hard rule was set, results discussed are those that are in the FDR -0.05 to -0.1 range.
- *Effect sizes*. Measured as fold change (FC) and calculated in R using Eqs. (2) and (3) for ANCOM-BC and MaAsLin2 respectively:

$$Fold\ change = \exp(x) \qquad (2)$$

$$Fold\ change = 2^{\wedge}x \qquad (3)$$

where $x$ is a vector of model coefficients derived from ANCOM-BC (based on natural log transformed abundances), or MaAsLin2 (based on log2-transformed relative abundances).

- *Plots*. The -log10 FDR q-values from MaAsLin2 and ANCOM-BC species MWAS were plotted for all tested species using ggplot2 R package. Venn diagrams showing overlap of MaAsLin2 and ANCOM-BC detected species from MWAS at FDR < 0.05 and FDR < 0.1 were created using the ggvenn R package. Relative abundances (MaAsLin2) and bias-corrected observed abundances (ANCOM-BC) in PD and NHC, and the FCs with 95% CI were plotted for PD-associated species and selected KO groups

and pathways using ggplot2 R package. To extract bias-corrected observed abundances from ANCOM-BC to plot, we used Eq. (4) in R:

$$Bias - corrected\ observed\ abundances = \exp(\log(x+1) - y) \qquad (4)$$

where $x$ is a sample by feature matrix of observed abundances, and $y$ is a vector of sampling fractions estimated by ANCOM-BC for each sample.

- *Parameters*. MaAsLin2 and ANCOM-BC were run with default parameters, except: microbial features that were present in <5% of samples were excluded from testing to ensure minimum analytic N=37 ('zero_cut=0.95' for ANCOM-BC, 'min_prevalence=0.05' for MaAsLin2), and multiple testing correction was Benjamini-Hochberg FDR[67] ('p_adj_method="BH"' for ANCOM-BC, 'correction="BH"' for MaAsLin2). Additionally, for MaAsLin2, 'normalization' was set to "NONE" as data input was already total sum scaled normalized, and 'standardize' was set to "FALSE" as standardization of quantitative variables was done prior to running MaAsLin2 using the 'scale' function in R.
- *Age and sex adjustment*. 84 PD-associated species from MWAS were retested for association with PD with age, sex, stool collection method, and total sequence count in the model, using MaAsLin2. ANCOM-BC was not used in subset analyses because bias-correction would use only the 84 species and not the entire metagenome. Benjamini-Hochberg FDR method[67] was used for multiple testing correction.
- *Confounder analysis*. To investigate the confounding effects of extrinsic variables that differed between PD and NHC, MaAsLin2 was performed with the 84 PD-associated species from MWAS, and 9 variables in the model: PD/NHC, total sequence count, intake (yes/no) of alcohol, laxatives, probiotics, antihistamines, depression-anxiety-mood medication, pain medication, and sleep-aid. Five subjects were excluded because they were missing pain medication and sleep-aid data. Stool collection method was not included as the 5 excluded subjects were those who used swabs. Although PD and NHC also differed in weight loss, GI discomfort, and constipation, these variables were not considered confounders because they are intrinsic features of PD. Adjusting for them would mask a part of PD, which is counter to the aim of this study: to generate a full unaltered view of the dysbiosis in PD metagenome. Age and sex are also intrinsic to PD as they are risk factors. However, the selection of household controls to minimize environmental variation inflated the female preponderance and younger ages in controls as PD is more prevalent in men. To ensure the species-level GWAS results were not artifacts of this distortion, we ran MaAsLin2 on the 84 PD-associated species from MWAS and 5 variables in the model: PD/NHC, total sequence count, stool collection method, male/female, and age at stool collection (in years and standardized using 'scale' in R). Benjamini-Hochberg FDR method[67] was used for multiple testing correction.

**Hypothesis-driven differential abundance analyses.** The following tests were conducted using linear regression (via 'lm' in R) on log2-transformed relative abundances as done with MaAsLin2. Each test was hypothesis driven, and no multiple testing was involved. Although tests had a specified direction, we present two-sided P-values to be conservative. Three hypotheses were tested. They were independent and unrelated; however, should one see appropriate to apply Bonferroni correction for 3 hypotheses, all three will retain significance at Bonferroni corrected, two-sided, P=4E-3 for elevated *Prevotella* sub-genus in PD vs. NHC, at P=6E-5 for elevated Cluster #17 in PD vs. NHC,

and for reduced sporulation KO and constipation at P=6E-4 in PD and at P=0.01 in NHC.

- *Prevotella* sub-genus. This test was conducted to assess reproducibility of an earlier finding of elevated abundances of a *Prevotella* sub-genus[17] that was identified by the 16S database SILVA, and included *Prevotella buccalis, timonensis, bivia, disiens* and *oralis*. At species-level they were individually too rare and excluded from MWAS. We collapsed relative abundances of these species to create the *Prevotella* sub-genus and tested the difference in PD vs. NHC.
- *Cluster #17*. This test was conducted to assess reproducibility of an earlier finding of elevated abundances of opportunistic pathogens detected at genus-level[17]. Again, individual species were rare and not tested, except *Porphyromonas asaccharolytica*. We combined them, as was done in prior study with genera, and tested the difference between PD and NHC. Test was performed once with all 19 species of cluster #17 combined (listed in Supplementary Data 8), and once excluding *Porphyromonas asaccharolytica*.
- *Sporulation KO groups and constipation*. This test was conducted because sporulation KO groups were reduced in PD, and prior studies have shown sporulation species modulate gut motility;[49] hence the question: Do our data support an association between reduced levels of sporulation KO groups and constipation? To test this, we collapsed the 27 sporulation KO groups that had shown association with PD (i.e., in Supplementary Data 9, searched KO group names for key word "sporulation", and selected those with FDR < 0.05 by ANCOM-BC and MaAsLin2), and tested their combined relative abundance in constipated (in past 3 months) vs. non-constipated strata, within PD and NHC separately. To determine if sporulation KO groups were significantly depleted in PD independently of constipation, we re-tested the association of case status with abundance of sporulation KO groups, this time adjusting for constipation in the model.

**Network analysis.** Correlation networks were generated for PD and NHC microbiomes separately. Pairwise SparCC correlations[68] were calculated using species count data as input to 'fastspar' from FastSpar (a C++ implementation of SparCC) specifying 100 iterations. Permuted *P*-values were then calculated for each correlation by (1) creating 1000 random datasets, (2) calculating SparCC correlations for each random dataset, and (3) calculating *P*-values for each correlation by determining the proportion of random correlations that were stronger than the original correlations. The Louvain algorithm for community detection[69] was used (via 'cluster_louvain' from the igraph R package) to algorithmically detect clusters of species using correlations that reached $|r| > 0.2$ and uncorrected permuted $P < 0.05$. The number of correlations at $|r| > 0.2$ and uncorrected permuted $P < 0.05$ for each species was calculated using 'degree' from igraph. To visualize networks, correlations that reached $|r| > 0.2$ with uncorrected permuted $P < 0.05$ were imported into Gephi along with their cluster memberships. The Force Atlas 2 algorithm[70] was then used to plot the network, showing species as nodes, correlations as connecting edges between species, and strength of correlation by the weight of the edge.

## Reproducibility

The results can be reproduced using raw sequence and metadata that are publicly available on NCBI SRA PRJNA834801, or using processed data (post-QC and taxonomic and functional profiling) in the "Source Data" on Zenodo [https://zenodo.org/record/7246185], following the Methods section of the manuscript and using the "Supplementary Code" provided with this manuscript, also on Zenodo [https://zenodo.org/record/7246185].

## Reporting summary

Further information on research design is available in the Nature Portfolio Reporting Summary linked to this article.

## Data availability

The datasets generated and analyzed during the current study are available to the public with no restrictions at the NCBI Sequence Read Archive (SRA) under BioProject ID PRJNA834801. Sequences on SRA are pre-QC. "Source Data", the post-QC and post taxonomic and functional profiling data generated here, is provided to the public with no restrictions at Zenodo [https://zenodo.org/record/7246185]. The human reference genome used for decontaminating metagenomic sequences (GRCh38.p13) is publicly available at NCBI under the GenBank assembly accession number GCA_000001405.28. The ChocoPhlAn database (v30) used by MetaPhlAn3 for taxonomic profiling is publicly available for download from the MetaPhlAn databases FTP site [http://cmprod1.cibio.unitn.it/biobakery3/metaphlan_databases] or by running the command 'metaphlan --install'. The ChocoPhlAn and UniRef90 reference databases (v201901b) used during functional profiling with HUMAnN3 are publicly available to download from the HUMAnN data FTP site [http://huttenhower.sph.harvard.edu/humann_data] or by using the 'humann_databases' utility script. Pathway data for functional profiling of pathways by HUMAnN come packaged with the HUMAnN program and were derived from the publicly accessible MetaCyc reference database (v24) [https://metacyc.org/].

## Code availability

Code used to perform sequence QC, taxonomic and functional profiling, statistical analyses, and figure generation are provided as "Supplementary Code" in the supplement of the manuscript and to the public with no restrictions on Zenodo [https://zenodo.org/record/7246185].

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

## Acknowledgements

The authors thank individuals who participated in this research project, and the movement disorder specialists at UAB. This work was supported by The U.S. Army Medical Research Materiel Command endorsed by the U.S. Army through the Parkinson's Research Program Investigator-Initiated Research Award under Award No. W81XWH1810508 (to H.P.); NIH Training Grant T32NS095775 (to Z.D.W.); NIH/NIEHS 1R01ES032440-01A1 and Parkinson's Foundation PF-SF-JFA-830658 (to T.S.); and Aligning Science Across Parkinson's [ASAP-020527] through the Michael J. Fox Foundation for Parkinson's Research (MJFF) (to H.P. and T.S.). For the purpose of open access, the authors have applied a CC BY public copyright license to all Author Accepted Manuscripts arising from this submission. Opinions, interpretations, conclusions, and recommendations are those of the authors and are not necessarily endorsed by the funding agencies.

## Author contributions

All authors met all four criteria (1) substantial contributions to the conception (H.P.) or design (H.P., Z.D.W.) of the work or the acquisition (H.P., D.G.S., M.N.D.), analysis (Z.D.W., G.C., G.T., A.D., H.P.) or interpretation of the data (H.P., T.S., Z.D.W.); (2) drafting the work (H.P., Z.D.W., T.S., A.D.) or revising it critically for important intellectual content (all authors); (3) final approval of the completed version (all authors); (4) accountability for all aspects of the work in ensuring that questions related to the accuracy or integrity of any part of the work are appropriately investigated and resolved (all authors).

## Competing interests

The authors declare no competing interests.
