## [Peer Review File · Nature Communications]

REVIEWER COMMENTS

Reviewer #1 (Remarks to the Author):

In their manuscript, the authors used deep shotgun sequencing of DNA extracted from faecal samples of 490 PD and 234 control individuals to identify microbial abundance and metabolic alterations in PD patients. The authors concluded the presence of microbial dysbiosis in PD patients, in line with previous reports, in addition to dysregulated neuroactive signaling, preponderance of molecules that induce alpha-synuclein pathology.

Major comments:

1. Page 4, "Human studies of PD and microbiome have had limited sample sizes and nearly all were based on 16S rRNA gene amplicon sequencing (henceforth, 16S) which limits resolution to genus-level.". This statement is not correct. The authors should check Genome Med. 2017;9(1):1–13.
2. Page 8, "We noted two clusters with species that are known to cause infections. Escherichia coli, Klebsiella pneumoniae and Klebsiella quasipneumoniae , which were elevated in PD, were positively correlated and grouped into cluster #6 of the PD network.". The authors label these bacteria as pathogenic. However, it should be noted that these species are typical small intestinal bacteria present in healthy subjects.
3. Page 8: "When taken together, the relative abundance of species in cluster #17 was significantly elevated in PD (fold change (FC)=2.63, P=2E-5). Is this significance after FDR correction? Please clarify. Likewise throughout the manuscript.
4. Page 9: "and (3) probiotics Bifidobacterium and Lactobacillus (elevated)". These genera have been reported in many studies to be elevated independent of the administration of probiotics nor the presence of PD.
5. "Alignment with existing literature". This section is missing a proper comparison with the existing study Genome Med. 2017;9(1):1–13.
6. Page 14: " Dopamine. A substantial drop in dopamine levels due to progressive loss of dopaminergic cellsto generate precursors to dopamine and possibly other catecholamines such as adrenaline.". This statement is not correct. TDC most likely compete with the uptake of Tyrosine in the tissue since it can be produced from Phe. Nonetheless, higher levels of TDC have been previously detected in PD patients to reduce the bioavailability of the levodopa treatment. All references in this regard are missing here.
7. Page 15 " Tryptophan, an essential amino acidsuggesting that serotonin production in the host is depleted." Again this link is not precise. Sporulating bacteria such as Clostridia induces

serotonin production from enterochromaffin cells (ECs). EC produced serotonin, unlike serotonin produced in the enteric nervous system, has a marginal role in regulating gut motility and cannot pass the BBB. Thus, even bacteria known to induce its production in the gut are lower in abundance in PD patients does not necessarily correlate with lower levels of brain serotonin. Moreover, 38% of the PD patients were under antidepressants, which would certainly elevate levels of serotonin in the brain.

8. Page 15: “ Given that spore forming bacteria stimulate gut motility..... in PD than in NHC (FC=0.79, P=5E-4). This conclusion is not precise. See my comment in point 7.

Reviewer #2 (Remarks to the Author):

The authors here present a large-scale metagenomics study of Parkinson’s disease (PD). This study is timely and, for the first time, provides a unique chance to uncover and confirm mixed results obtained from earlier studies due to limited sample sizes. The organization of the manuscript is clear, the scientific interpretations are comprehensive, and the efforts to replicate the validate findings from other studies are highly appreciated. Since the clinical and scientific relevance are clearly stated, below are some comments mainly focus on statistical methods and interpretations that I hope could help further streamline the text.

General comments:

1. Sample size.

1a. Lines 171-173. Authors stated that 697 and 499 species were used to calculated SparCC. Please explain the data attribution criterium here, and perhaps state it in Fig. 1.

1b. Lines 595-596. Authors stated there are 487 PD and 232 NHC for confounding analysis. Please explain how to get these numbers, and perhaps state it in Fig. 1.

2. Beta diversity.

2a. Lines 617-619. The overall difference was tested using PERMANOVA. Permutational Analysis of Multivariate Dispersion (PERMDISP) can be added to determine the leading effect of a significant PERMANOVA result (differential spatial medians or heterogeneity of dispersions).

2b. Supplementary Fig. 2. PC1 and PC2 together explain less than 14% of the total variance, which means there is a higher portion of un-structured technical variation in the system. It is not uncommon in microbiome or sc-RNA studies due to the inherent sparsity and heterogeneity; however, people usually implement a feature selection step before the dimension reduction (for example, see here: https://satijalab.org/seurat/articles/pbmc3k_tutorial.html). I wonder if a supplementary analysis could be added to test how robust the result is by excluding rare taxa (e.g. prevalence < 5%).

3. Differential abundance analysis.

3a. Lines 595-600. It seems the library size (total sequence count) was included for all differential abundance analyses. Although it is fine to adjust it for relative

abundance analysis as shallow sequencing depth could affect rare taxa and thus change the whole microbial composition (please also state in the corresponding section if there are other reasons), including this term in absolute abundance analysis seems overfitting. For ANCOM-BC model, its sampling fraction term has

already taken into account the library size and thus adjusting it one more time could diminish the power.

3b. Lines 644-647. Why were the significance level set differently for ANCOM-BC and MaAsLin2? Which one is 0.05, and which one is 0.1?

3c. Could authors draw a Venn Diagram comparing results between relative abundance analysis (MaAsLin2) and absolute abundance analysis (ANCOM-BC)? It would be easier to see how consistent (or different) the results are by a Venn Diagram.

4. Code availability

The code repository is not publicly available while requesting the access might break anonymization. Is there a way to generate a link specifically for reviewers?

Minor comments:

1. Supplementary Fig. 3. Use "bias-corrected abundances" for the title of column b.

2. Supplementary Tables were cut off and hard to see

3. Line 161. Authors stated that the results are the same after adding age and sex in the model. I might have missed it but was not able to locate which table or figure contains this information.

4. Fig. 2 is not easy to read. Please consider making some sub-figures by zooming in on certain regions (e.g. cluster #2) and labeling some important species/genera.

Reviewer #3 (Remarks to the Author):

Wallen et al investigated the gut microbiome of individuals with PD and controls, using shotgun metagenomics, association analysis and functional annotations. They observed key changes on the compositional and functional changes in the gut microbiome of PD individuals. Performing the network analysis and clustering the species in cooperation and competition were identified. These outcomes later linked to the inflammatory and neuroprotective factors through bacterial toxin productions. While I appreciate the importance of the microbiome in the PD and new shotgun metagenome sequencing of the cohort, I found the results of the manuscript stays on the species level, co-abundance and network analysis. The key part of the paper was the functional and pathway annotations, which some of these results have reported before and the authors also referred to them in their work. The pathway and functional analysis remain in a statistical level, while they are testable hypothesis and to raise the impact of the findings and this work I see it necessary to at least test some of their new findings.

Here also some other specific comments:

The identified species mentioned was total of 2270. Is this the total number of available species in the MetaPhlan3? How did this number recover?

As the depth of the sequence are varied, any normalization or downsizing has been applied on the samples before downstream analysis?

The 719 species that was selected out of 2270 through their QC; however I didn't find the method, threshold and rationale behind this selection. This is quite important step of the analysis and proper explanation in the main text and method would be necessary.

Also just 257 species present in more than 5% of the populations, how comparative is this number to other studies? Moreover, what was the presence/absence criteria here? Was abundance value used and if yes what threshold?

Table 2 is useful to have the entire list of species, however I recommend to present an overview figure and visualization for the taxonomy differences between the PD and NHC in the main figures.

Additionally the first part of results, before the network analysis, I didn't observed any report on the key species and genera. I see it necessary to mention the key findings at this stage.

-Introducing microbiome and microbiota may not be useful as these are well known terms.

-L52-54 lacks the relevant references for studies in the diseases associated with microbiome.

-L56-57 -> reference?

-L89 -> Would be good to refer to "Functional implications of microbial and viral gut metagenome changes in early stage L-DOPA-naïve Parkinson's disease patients.

Genome Med. 2017; 9: 39" as one of the first metagenomics paper on PD

-It could be useful to have the last paragraph on overall of the manuscripts rather than details about 16S and metagenomics advantageous and disadvantageous.

L109 – 110 -> We couldn't access the raw sequences and it was not public at the time of reviewing.

L110-111 -> I am confused about the age match, as elsewhere in the ms mentioned all the individuals in this work are above 50!

L135-136 -> How the one value of significance was calculated for enterotype and please provide the type of the test in the text. Also for the enrichment of Firmicute what is the stats and value?

L139 -> Not entirely agree it is just species. One of the great applications of shotgun data is as well gene and functional annotation for further mechanism analysis.

I suggest to revise the results subtitle and refer to the main outcome of the section. As it is stands like "alignment with existing literature" or "replication" doesn't really provide informative information of the section.

POINT-BY-POINT RESPONSE TO REVIEWS

REVIEWER COMMENTS

Reviewer #1 (Remarks to the Author):

In their manuscript, the authors used deep shotgun sequencing of DNA extracted from faecal samples of 490 PD and 234 control individuals to identify microbial abundance and metabolic alterations in PD patients. The authors concluded the presence of microbial dysbiosis in PD patients, in line with previous reports, in addition to dysregulated neuroactive signaling, preponderance of molecules that induce alpha-synuclein pathology.

Major comments:

1. Page 4, "Human studies of PD and microbiome have had limited sample sizes and nearly all were based on 16S rRNA gene amplicon sequencing (henceforth, 16S) which limits resolution to genus-level.". This statement is not correct. The authors should check Genome Med. 2017;9(1):1–13.

The statement is correct but as reviewer points out, it can be more precise by changing "nearly all" to "all except two^{22,23}" and citing Bedarf et al (Genome Med. 2017;9(1):1–13) as reviewer requested, as well as Qian et al. (Brain 2020). Line 87.

2. Page 8, "We noted two clusters with species that are known to cause infections. Escherichia coli, Klebsiella pneumoniae and Klebsiella quasipneumoniae , which were elevated in PD, were positively correlated and grouped into cluster #6 of the PD network.". The authors label these bacteria as pathogenic. However, it should be noted that these species are typical small intestinal bacteria present in healthy subjects.

We added on lines 230-232:

"These species are commensal to gut microbiome, as evinced by their presence in NHC, but have the capacity to become opportunistic pathogens."

3. Page 8: "When taken together, the relative abundance of species in cluster #17 was significantly elevated in PD (fold change (FC)=2.63, P=2E-5). Is this significance after FDR correction? Please clarify. Likewise throughout the manuscript.

About Cluster 17: The test was hypothesis-driven based on prior results from 16S datasets, it was a single test, and did not involve multiple tests to apply FDR to. To clarify, and, to preemptively address other questions that may occur to the reviewers and the readers, we expanded the section "5.3.2. Hypothesis-driven differential abundance analyses" by adding on line 760-765:

"Each test was hypothesis driven, and no multiple testing was involved. Although tests had a pre-specified direction, we present two-sided P value to be conservative. Three hypotheses were tested. They were independent and unrelated; however, should one see appropriate to apply Bonferroni correction for 3 hypotheses, all three will retain significance at Bonferroni

corrected, two-sided, $P=4E-3$ for elevated *Prevotella* subgenus in PD vs. NHC, at $P=6E-5$ for elevated Cluster #17 in PD vs. NHC, and for reduced sporulation KO and constipation at $P=6E-4$ in PD and at $P=0.01$ in NHC.”

In addition, in Fig. 1, we added “hypothesis-based” to every test that is hypothesis-based.

Regarding throughout the manuscript: We organized the statistical significance approach (FDR, P value and rationale for/against multiple testing correction) in one place (Methods / 5. Statistical testing) rather than having to mention it repeatedly with each piece of data throughout the manuscript. We would like to keep this organization. Please see Methods / 5. Statistical analysis, where we added on lines 653-654:

“Significance determination was based on the test, whether agnostic or hypothesis-driven, and corrected for multiple testing accordingly, as noted below for each analysis.”

The significance for each test appears on lines 660-661 (metadata), 673-674 (PERMANOVA), 677-678 (PERMDISP), 680-684 (enterotype), 705-711 (MWAS), 739-740 (MWAS adjusted for sex and age), 756-757 (confounder adjustment), 760-765 (hypothesis-based), 790-793 (correlations).

1. Page 9: “and (3) probiotics Bifidobacterium and Lactobacillus (elevated).”. These genera have been reported in many studies to be elevated independent of the administration of probiotics nor the presence of PD.

If the objection is to the use of the word probiotic, we deleted it.

If the point is that elevated *Lactobacillus* and *Bifidobacteria* have been reported in non-PD studies, we agree. Same goes for SCFA producing taxa, which are reduced in many inflammatory disorders. The reviewer’s comment raises an excellent point about specificity of these associations, worth noting. We added this concluding remark on lines 283-285: “It is not known which of these changes are specific to PD. Elevated *Bifidobacterium* and *Lactobacillus* and loss of SCFA producing bacteria have also been observed in other inflammatory disorders of the gut³¹.”

3. “Alignment with existing literature”. This section is missing a proper comparison with the existing study Genome Med. 2017;9(1):1–13.

There are two PD studies with shotgun data. We added a comparison of our study with both Genome Med. 2017;9(1):1–13. and Qian et al Brain 2020. Lines 296-307.

“Only two prior PD studies used shotgun sequencing, conducted by Bedarf et al.²¹ and Qian et al.²². Compared to Bedarf et al.²¹ we differed in sample size ($N=59$ vs. $N=724$), sex (100% male vs. 52% male), patient characteristics such as constipation (not relevant vs. highly significant), geography (Bonn, EU vs. Deep South, US), and taxonomic profiling (MOCAT2 vs. MetaPhlan3), to name a few of a myriad of factors that can have profound effects on results. We confirmed their original report of reduction in *Prevotella copri* and alterations in tryptophan metabolism. They also reported reduction in total virus abundance in PD. We have not yet investigated the virome because virus databases are still sparse and detection methods underperform³². Qian et al.’s study²² also differed from our study in many aspects, most notably in race (Chinese vs. Caucasian) and study design (gene-based machine learning classifier for prediction vs. delineation of alterations at global, species, gene, and pathway level and functional inference). Despite differences, we confirmed their report that majority of species enriched in PD were in *Firmicutes* phylum, and *Alistipes* of *Bacteroidetes* phylum was highly enriched in PD.”

4. Page 14: “Dopamine. A substantial drop in dopamine levels due to progressive loss of dopaminergic cellsto generate precursors to dopamine and possibly other catecholamines such as adrenaline.” This statement is not correct. TDC most likely compete with the uptake of Tyrosine in the tissue since it can be produced from Phe. Nonetheless, higher levels of TDC have been previously detected in PD patients to reduce the bioavailability of the levodopa treatment. All references in this regard are missing here.

About the statement being incorrect: I have pasted the entire section (lines 424-432), and will address it point-by-point:

“A substantial drop in dopamine levels due to progressive loss of dopaminergic cells in the brain is the defining feature of PD This is the classic definition. We detected elevated levels of tyrosine decarboxylase/aspartate 1-decarboxylase gene-family (K18933: FC=1.57, FDR=3E-4) This is data which removes tyrosine This is correct, a required precursor of dopamine This is correct, thus limits dopamine production by host and bacteria This is inference, we modified it to “Thus, one may infer that elevated K18933 could limit dopamine production by host and bacteria. The aromatic amino acid synthesis pathway (COMPLETE-ARO-PWY: MaAsLin2 FC=0.95, FDR=6E-3; ANCOM-BC FC=0.74, FDR=1E-8) was depleted in PD metagenome This is data, suggesting that the microbiome has diminished capacity to generate precursors to dopamine and possibly other catecholamines such as adrenaline. This is clearly inference as it is said to be suggestive.”

About TDC: We can do better than just cite references; we have results. We had not included *tdc* in the Functional Inference section because inferences are based on genes and pathways, and we did not detect the *tdc* gene or TDC pathway. But we did detect species that code for them and they are elevated in PD. This was added on lines 424-432:

“Tyrosine decarboxylase (TDC) is noteworthy because it is a bacterial enzyme capable of metabolizing L-dopa to dopamine in the gut before it can reach the brain. TDC is encoded by bacterial *tdc* gene. *E. faecium* and *E. faecalis* *tdc* genes metabolize L-dopa efficiently, and their abundance in the gut correlates directly with reduced efficacy of L-dopa for the patient^{43,44}. We did not detect the *tdc* gene or the TDC pathway in our dataset, not necessarily because they do not exist but more likely that they were filtered out by the stringent QC criteria. We did however detect the species, *E. faecium* and *E. faecalis*, and both were elevated in PD. *E. faecium* rose to significance in MWAS and was nominated as PD-associated species with FC=1.4 by MaAsLin2 and FC=1.8 by ANCOM-BC, both with FDR~0.02. *E. faecalis* yielded FC=1.2 with FDR=0.1 by MaAsLin2 and FC=1.3 with FDR=0.2 by ANCOM-BC.”

5. Page 15 “Tryptophan, an essential amino acidsuggesting that serotonin production in the host is depleted.” Again this link is not precise. Sporulating bacteria such as Clostridia induces serotonin production from enterochromaffin cells (ECs). EC produced serotonin, unlike serotonin produced in the enteric nervous system, has a marginal role in regulating gut motility and cannot pass the BBB. Thus, even bacteria known to induce its production in the gut are lower in abundance in PD patients does not necessarily correlate with lower levels of brain serotonin. Moreover, 38% of the PD patients were under antidepressants, which would certainly elevate levels of serotonin in the brain.

About marginal role in gut motility: Even at marginal, it is measurable and necessary, as reported in the literature (PMID: 25860609), and our findings here show that association of sporulation KO abundance with constipation is detectable at metagenome level, with notable effect sizes of about 30%-70% reduction and high statistical significance (FDR=0.0002 in PD, FDR=.003 in NHC). Lines 455-459.

About passing the BBB, and serotonin levels in the brain: We make no claim about serotonin and the brain. The focus here is on intestinal serotonin stimulated by gut bacteria and what it may do in the gut and to the gut – we know it affects gut motility which is a problem in PD. To prevent readers from assuming we are making a link to the brain, we revised the last sentence by inserting “intestinal” and making it double suggestive: “suggesting that intestinal serotonin production in the host may be diminished.” Line 451.

8. Page 15: “ Given that spore forming bacteria stimulate gut motility in PD than in NHC (FC=0.79, P=5E-4). This conclusion is not precise. See my comment in point 7.

We have pasted the section in full (lines 453-462), and address it point-by-point:

“Given that spore forming bacteria stimulate gut motility⁵⁰ Evidence based and cited, and our data suggest a global depletion of the sporulation KO groups This is data, we speculated, and tested the hypothesis that constipation, a common symptom of PD, may be related to the depletion of spore forming bacteria Hypothesis. We selected 27 sporulation KO groups that were FDR<0.05 in both ANCOM-BC and MaAsLin2 and tested the hypothesis that the relative abundance of sporulation KO groups is even lower in subjects who reported constipation vs. those that did not Test is outlined precisely. Indeed, lower abundances of sporulation KO groups was associated with constipation both in PD group (FC=0.76, P=2E-4), and in NHC group (FC=0.58, FDR=3E-3) This is data.”

In short, we outlined a hypothesis that we derived from data, then tested it, and show the results. There is no conclusion that would not be precise. We then questioned the robustness of our original finding of association of sporulation KO with PD and the possibility that it was secondary to constipation: We should note that association of sporulation KO groups with PD is not solely due to constipation; adjusting for constipation, the relative abundance of sporulation KO groups was still significantly lower in PD than in NHC (FC=0.79, P=5E-4) This is data.

Reviewer #2 (Remarks to the Author):

The authors here present a large-scale metagenomics study of Parkinson’s disease (PD). This study is timely and, for the first time, provides a unique chance to uncover and confirm mixed results obtained from earlier studies due to limited sample sizes. The organization of the manuscript is clear, the scientific interpretations are comprehensive, and the efforts to replicate the validate findings from other studies are highly appreciated. Since the clinical and scientific relevance are clearly stated, below are some comments mainly focus on statistical methods and interpretations that I hope could help further streamline the text.

General comments:

1. Sample size.

1a. Lines 171-173. Authors stated that 697 and 499 species were used to calculate SparCC. Please explain the data attribution criterion here, and perhaps state it in Fig. 1.

SparCC was conducted on PD and NHC separately. The numbers represent all species that were detected in each group: 697 in PD and 499 in NHC. They do not match the total 719 species because some species were detected in PD and not NHC or vice versa. More species were detected in PD because PD had larger sample size. There was no criterion to specify, we used them all. The numbers are in Fig. 1. We modified the sentence to make it clearer (lines 201-202):

“We calculated pairwise correlation (r) in species abundances for PD metagenome using all 697 species detected in PD (Supplementary Data 5), and for NHC metagenome using all 499 species detected in NHC (Supplementary Data 6).”

1b. Lines 595-596. Authors stated there are 487 PD and 232 NHC for confounding analysis. Please explain how to get these numbers, and perhaps state it in Fig. 1.

We apologize, that was a typographic transposition. The correct numbers for confounding analysis are 435 PD and 219 NHC. Text was corrected (line 649). Numbers were added to Table 2 and Fig. 1. These numbers can be derived from “Source Data” file, tab “subject_metadata”.

2. Beta diversity.

2a. Lines 617-619. The overall difference was tested using PERMANOVA. Permutational Analysis of Multivariate Dispersion (PERMDISP) can be added to determine the leading effect of a significant PERMANOVA result (differential spatial medians or heterogeneity of dispersions).

Thank you very much for this lead. We ran PERMDISP analysis, and it was significant. I wonder if it reflects PD heterogeneity, which might help us with subtyping based on microbiome (not part of this paper). Thanks again! The following were added:

Results lines 140-147:

“Test of dispersion was also significant ($P < 1E-4$, tested using permutational analysis of multivariate dispersion (PERMDISP)) indicating that the difference in global composition of PD vs. NHC is driven primarily by differences in dispersion, rather than differences in spatial medians. Principal component analysis (PCA) plots also show that PD metagenomes were visibly more dispersed than NHC (Supplementary Fig. 2 a,b). Results were robust when rare taxa (prevalence $< 5\%$) were excluded, yielding $P < 1E-4$ for both PERMANOVA and PERMDISP. Greater dispersion across PD metagenomes may reflect the heterogeneity of PD. PD is not one disease, and each disease mechanism may have a different microbiome signature”.

Methods lines 675-678:

“**5.2.3 Dispersion.** To test homogeneity of dispersion (the assumption in PERMANOVA that groups being compared (PD vs. NHC) are similar in dispersion) we performed PERMDISP using betadisper and permutest functions from the vegan R package. Significance of PERMDISP was determined using 9,999 permutations, which is capped at $P < 1E-4$ for the highest significance.”

2b. Supplementary Fig. 2. PC1 and PC2 together explain less than 14% of the total variance, which means there is a higher portion of un-structured technical variation in the system. It is not uncommon in microbiome or sc-RNA studies due to the inherent sparsity and heterogeneity; however, people usually implement a feature selection step before the dimension reduction (for example, see here: https://satijalab.org/seurat/articles/pbmc3k_tutorial.html). I wonder if a supplementary analysis could be added to test how robust the result is by excluding rare taxa (e.g. prevalence < 5%).

We ran the analyses as recommended. We have not seen seurat applied to microbiome, and the default being 2000 features is too high for microbiome. We specified 250 features which explained 16% of variation, and 100 features which explained 20% of variation. After removing rare species (prevalence <5%) the first two PCs together explained 14.3% of variance. We ran PERMANOVA and PERMDISP with every iteration (feature selection at 250, at 100, and with species with prevalence >5%), and results for all iterations were robustly significant in both PERMANOVA and PERMDISP, all staying at the $P < 1E-4$ (highest attainable with 9999 permutation) except PERMDISP for 100 feature selection was $P = 0.01$.

As recommended, the PCA with species at prevalence >5% was added as supplementary analysis to Supplementary Fig 2, as new panel b and to Results on lines 144-145:

“Results were robust when rare taxa (prevalence <5%) were excluded, yielding $P < 1E-4$ for both PERMANOVA and PERDISP.”

3. Differential abundance analysis.

3a. Lines 595-600. It seems the library size (total sequence count) was included for all differential abundance analyses. Although it is fine to adjust it for relative abundance analysis as shallow sequencing depth could affect rare taxa and thus change the whole microbial composition (please also state in the corresponding section if there are other reasons), including this term in absolute abundance analysis seems overfitting. For ANCOM-BC model, its sampling fraction term has already taken into account the library size and thus adjusting it one more time could diminish the power.

As suggested, we ran ANCOM-BC without sequence count in the model. See **Table R1** below for a side-by-side comparison of ANCOM-BC results with vs. without sequence count in the model. Results were strikingly similar, with no notable loss of power: Every species that was called significant with sequence count adjustment was also significant after removing sequence count covariate, and no new signals emerged after removing sequence count from the model.

3b. Lines 644-647. Why were the significance level set differently for ANCOM-BC and MaAsLin2? Which one is 0.05, and which one is 0.1?

Sorry for the confusion. We now clarify in Methods by adding “i.e.,” line 708-710:

“we set statistical significance for association with PD as concordance between ANCOM-BC and MaAsLin2 with $FDR < 0.05$ by one and $FDR \leq 0.1$ by the other (i.e., a taxon was called significance if it achieved $FDR < 0.05$ by both MaAsLin2 and ANCOM-BC, or $FDR < 0.05$ in MaAsLin2 and $FDR \leq 0.1$ in ANCOM-BC, or $FDR < 0.05$ in ANCOM-BC and $FDR \leq 0.1$ in MaAsLin).”

3c. Could authors draw a Venn Diagram comparing results between relative abundance analysis (MaAsLin2) and absolute abundance analysis (ANCOM-BC)? It would be easier to see how consistent (or different) the results are by a Venn Diagram.

We have a new Figure 2 with one Venn diagram for overlaps at $FDR < 0.05$ and one for overlaps at $FDR < 0.1$. Reviewer 3 suggested we should display species level results graphically in the main text, so we took the Venn diagram idea a step further and plotted the entire FDR distribution by MaAsLin2 vs. ANCOM-BC. We think it complements the Venn diagrams quite nicely by showing granular data for each species with the most notable species labeled, it displays species-level concordance between ANCOM-BC and MaAsLin, and it shows how high the FDRs go. It is summarized in Results on lines 168-170:

“By this stringent definition, 84 of 257 species tested were associated with PD: 64 achieved $FDR < 0.05$ by both MaAsLin2 and ANCOM-BC, 10 had $FDR < 0.05$ by MaAsLin2 and $FDR \leq 0.1$ by ANCOM-BC, and 6 had $FDR < 0.05$ by ANCOM-BC and $FDR \leq 0.1$ by MaAsLin2 (Figure 2).” And in Methods on lines 718-721:

“The $-\log_{10}$ FDR q-values from MaAsLin2 and ANCOM-BC species MWAS were plotted for all tested species using ggplot2 R package. Venn diagrams showing overlap of MaAsLin2 and ANCOM-BC detected species from MWAS at $FDR < 0.05$ and $FDR < 0.1$ were created using the ggvenn R package.”

4. Code availability

The code repository is not publicly available while requesting the access might break anonymization. Is there a way to generate a link specifically for reviewers?

Of course. The easiest way to share was to add it to the revised manuscript as a supplement. It is provided and called “Supplementary Code”.

Minor comments:

1. Supplementary Fig. 3. Use “bias-corrected abundances” for the title of column b.

Done.

2. Supplementary Tables were cut off and hard to see

Due to their large sizes, Supplementary Tables are now labeled as Supplementary Data and provided in Excel. Hopefully this alleviates the problem of being cut-off.

3. Line 161. Authors stated that the results are the same after adding age and sex in the model. I might have missed it but was not able to locate which table or figure contains this information.

That was an omission on our part. Please see new Supplementary Table 3 for age and sex adjusted MWAS results.

4. Fig. 2 is not easy to read. Please consider making some sub-figures by zooming in on certain regions (e.g. cluster #2) and labeling some important species/genera.

Three panels were added to Fig 2 to show zoomed in with the key species labeled in cluster 17, cluster 2, and clusters 8, 13 and 6.

Reviewer #3 (Remarks to the Author):

Wallen et al investigated the gut microbiome of individuals with PD and controls, using shotgun metagenomics, association analysis and functional annotations. They observed key changes on the compositional and functional changes in the gut microbiome of PD individuals. Performing the network analysis and clustering the species in cooperation and competition were identified. These outcomes later linked to the inflammatory and neuroprotective factors through bacterial toxin productions. While I appreciate the importance of the microbiome in the PD and new shotgun metagenome sequencing of the cohort, I found the results of the manuscript stays on the species level, co-abundance and network analysis. The key part of the paper was the functional and pathway annotations, which some of these results have reported before and the authors also referred to them in their work. The pathway and functional analysis remain in a statistical level, while they are testable hypothesis and to raise the impact of the findings and this work I see it necessary to at least test some of their new findings.

Thank you, indeed there is a treasure trove of hypotheses to test, which we are tackling methodically. In fact, we recently received a 9 million dollar 3-year ASAP/MJFF grant to do exactly what the reviewer deems necessary. We will be testing findings from human studies (statistics/this paper specifically) in gnotobiotic and humanized a-syn transgenic mice, and gut-derived organoids. This is a huge project to be done right. It is beyond the scope of this paper. That said, some of our findings have already been validated experimentally. Please see Functional Inference section where we state on line 340-343 “Most functional insights to PD pathogenesis are derived from model organisms and in experimental setting, with the caveat that their relevance to human PD is not guaranteed. Here, we provide data from the human PD gut metagenome that corroborate, and validate, some of the basic science findings.” For example: curli’s ability to induce a-syn aggregation, and that infection in the gut induces pathology in *PINK1*^{-/-} mice were discovered in animal models, and are being corroborated for the first time in human PD in this dataset.

Here also some other specific comments:

The identified species mentioned was total of 2270. Is this the total number of available species in the MetaPhlAn3? How did this number recover?

MetaPhlAn3 database has over 10,000 species. 2270 is the total number of species in our dataset that mapped to at least 1 marker gene in MetaPhlAn3. 2270 was then reduced to 719 for analysis by default QC parameters of MetaPhlAn3, as was stated in the Methods. We now list the default parameters specifically (Lines 629-631):

“(default parameters: t=rel_ab, perc_nonzero=0.33, stat_q=0.2, avoid_disqm=FALSE, stat=tavg_g, min_cu_len=2000, unknown_estimation=FALSE).”

We also modified the main text, adding more detail (lines 157-162):

“Including Bacteria, Archaea and Eukarya, we identified a total of 2,270 species that mapped to at least one marker gene, 719 of which passed stringent bioinformatic quality control (QC) thresholds of MetaPhlAn3 (default parameters: `t= rel_ab`, `perc_nonzero=0.33`, `statq=0.2`, `avoid_disqm=FALSE`, `stat=tavg_g`, `min_cu_len=2000`, `unknown_estimation=FALSE`). 257 species were present in >5% of subjects, i.e., 35% which is closely comparable to another large study²⁵.”

As the depth of the sequence are varied, any normalization or downsizing has been applied on the samples before downstream analysis?

We did not downsize or rarify (PMID: 24699258). We used the following steps to account for variation in sequencing depth: In MetaPhlAn3, relative abundances were normalized to the total sequence count per sample. In ANCOM-BC, bias-corrected abundance estimates take total sequence count per sample into account. We used center log ratio (clr) transformation of abundances for PCA, PERMANOVA, and PERMDISP, and log-ratios of abundances for SparCC. We included total sequence count per sample as covariate and adjusted for it in all statistical analyses. These are specified in Methods.

The 719 species that was selected out of 2270 through their QC; however I didn't find the method, threshold and rational behind this selection. This is quite important step of the analysis and proper explanation in the main text and method would be necessary.

Yes, they are important. We added the necessary detail to the main text (lines 157-162):

“Including Bacteria, Archaea and Eukarya, we identified a total of 2,270 species that mapped to at least one marker gene, 719 of which passed stringent bioinformatic quality control (QC) thresholds of MetaPhlAn3 (default parameters: `t= rel_ab`, `perc_nonzero=0.33`, `statq=0.2`, `avoid_disqm=FALSE`, `stat=tavg_g`, `min_cu_len=2000`, `unknown_estimation=FALSE`). 257 species were present in >5% of subjects, i.e., 35% which is closely comparable to another large study²⁵.”

And repeated the default parameters in Methods (Lines 629-631):

“(default parameters: `t= rel_ab`, `perc_nonzero=0.33`, `statq=0.2`, `avoid_disqm=FALSE`, `stat=tavg_g`, `min_cu_len=2000`, `unknown_estimation=FALSE`)”

Also just 257 species present in more than 5% of the populations, how comparative is this number to other studies? Moreover, what was the presence/absence criteria here? Was abundance value used and if yes what threshold?

It is comparable to the large study conducted by the developers of MetaPhlAn3 themselves (PMID: 33944776, we used data in their supplementary file 5 to generate these numbers). In their study 917 species passed MetaPhlAn QC, and 303 of them (33%) had prevalence >0.05. In our study 719 species passed MetaPhlAn QC and 257 (35%) had prevalence >0.05. Note that they identified more species from the start because they had larger sample size (N=1266 metagenomes from 17 datasets combined vs. N=724 metagenomes in our one dataset).

The presence/absence criterion was relative abundance greater than zero. No abundance threshold was used for presence/absence. We added the following to the main text on line 161-162:

“257 species were present in >5% of subjects, i.e., 35% which is closely comparable to another large study²⁵.”

Table 2 is useful to have the entire list of species, however I recommend to present an overview figure and visualization for the taxonomy differences between the PD and NHC in the main figures.

We have added a new Fig 3 to address this comment specifically. In response to another reviewer, we added Fig 2, which further complements Fig. 3 in giving an overview of the differences between PD and NHC.

Additionally the first part of results, before the network analysis, I didn't observed any report on the key species and genera. I see it necessary to mention the key findings at this stage.

In response to three separate but related comments from reviewers, we have added new Figures 2 and 3 and expanded the Results:

Lines 168-170: “By this stringent definition, 84 of 257 species tested were associated with PD: 64 achieved $FDR < 0.05$ by both MaAsLin2 and ANCOM-BC, 10 had $FDR < 0.05$ by MaAsLin2 and $FDR \leq 0.1$ by ANCOM-BC, and 6 had $FDR < 0.05$ by ANCOM-BC and $FDR \leq 0.1$ by MaAsLin2 (Fig. 2).

Lines 176-180 “Effect sizes were large (Fig. 3). At one end of the spectrum, *Bifidobacterium dentium* was elevated by 7-fold, *Actinomyces oris* by 6.5-fold, *Streptococcus mutans* by 6-fold. At the other end of the spectrum, *Roseburia intestinalis* was reduced by 7.5-fold, and *Blautia wexlerae* by 5-fold. Overall, 36% (30 of 84) of PD-associated species had higher than two-fold change in abundance by both MaAsLin2 and ANCOM-BC, reflecting a 100% to 750% increase or decrease in PD vs. NHC.”

-Introducing microbiome and microbiota may not be useful as these are well known terms.

Deleted the definitions.

-L52-54 lacks the relevant references for studies in the diseases associated with microbiome.

Line 52: We did have one reference, a review (Lynch 2016, NEJM), which we cut and replaced with another review (Fan 2021, Nat Rev Microbiol) which we are already using, in order to keep number of references at 70 max allowed. Our paper covers a lot of topics and we had to be frugal with references, which meant using and reusing mostly reviews for general topics and saving original articles for specific topics. With the additions made in this revision, we had to cut and consolidate ~20 references to stay at 70. That said, if reviewer feels more references are

necessary for diseases associated with microbiome, and the editor permits us, we could add more: PMID:36004400, 31910983, 29634682.

-L56-57 -> reference?

Line 55. Added the reference to Dorsey 2018.

-L89 -> Would be good to refer to “Functional implications of microbial and viral gut metagenome changes in early stage L-DOPA-naïve Parkinson’s disease patients. *Genome Med.* 2017; 9: 39” as one of the first metagenomics paper on PD

Reviewer 1 also recommended we include *Genome Med.* 2017; 9: 39 paper and do a “proper comparison” to our study. There are in fact two shotgun studies in PD, if we single out one, we should also include the other. This was added to text: Lines 296-307

Only two prior PD studies used shotgun sequencing, conducted by Bedarf et al.²¹ and Qian et al.²². Compared to Bedarf et al.²¹ we differed in sample size (N=59 vs. N=724), sex (100% male vs. 52% male), patient characteristics such as constipation (not relevant vs. highly significant), geography (Bonn, EU vs. Deep South, US), and taxonomic profiling (MOCAT2 vs. MetaPhlan3), to name a few of a myriad of factors that can have profound effects on results. We confirmed their original report of reduction in *Prevotella copri* and alterations in tryptophan metabolism. They also reported reduction in total virus abundance in PD. We have not yet investigated the virome because virus databases are still sparse and detection methods underperform³². Qian et al.’s study²² also differed from our study in many aspects, most notably in race (Chinese vs. Caucasian) and study design (gene-based machine learning classifier for prediction vs. delineation of alterations at global, species, gene, and pathway level and functional inference). Despite differences, we confirmed their report that majority of species enriched in PD were in *Firmicutes* phylum, and *Alistipes* of *Bacteroidetes* phylum was highly enriched in PD.

-It could be useful to have the last paragraph on overall of the manuscripts rather than details about 16S and metagenomics advantageous and disadvantageous.

We fixed it by moving the first three sentences of last paragraph to the end of the prior paragraph (Lines 87-91). Now the last paragraph is on overall of the manuscript and we added general findings per Nat Commons formatting instructions (Lines 93-100):

“Here we present a large-scale metagenomics analysis of PD gut microbiome. This study was designed and executed by a single team of investigators (NeuroGenetics Research Consortium, NGRC), enabling complete control to employ state of the art methods and ensure uniformity from start to end. We confirm the common findings from prior studies, resolve them to species level and solve the inconsistencies in the literature. In addition, owing to large sample size and deep shotgun sequencing, we generate a vast amount of new information. We show widespread dysbiosis in PD microbiome, identify species that drive the dysbiosis, and by functional profiling, nominate microbial genes and pathways in the gut that may contribute to PD mechanisms.”

L109 – 110 -> We couldn't access the raw sequences and it was not public at the time of reviewing.

We reached out to SRA, but they do not allow access, not even to us, before release. We have told the editor that we are prepared to share our copy via Box. Just let the editor know and we will send them the link which can be distributed to you anonymously. We are also providing a new file with this revision called "Source Data", which contains all the data necessary to reproduce all the figures and tables. "Source data" is post-QC and post taxonomic and functional profiling, whereas raw sequences are pre-QC.

L110-111 -> I am confused about the age match, as elsewhere in the ms mentioned all the individuals in this work are above 50!

The way we had compared our dataset to HMP (who are younger) made it confusing. We rewrote this section (Lines 107-112)

"The sample size is comparable to Human Microbiome Project (HMP) which included 100 individuals with inflammatory bowel disease, 106 individuals with pre-diabetes and 242 healthy adults ages 18-40 years old²⁴. Defined by self-reported biological sex, 52% of subjects were men, 48% were women. 97% of PD cases and 93% of NHC were over 50 years old. The older ages of the controls in this study (mean 65.8±8.8) and their neurologically healthy status is a unique addition to the publicly available datasets."

L135-136 -> How the one value of significance was calculated for enterotype and please provide the type of the test in the text. Also for the enrichment of Firmicute what is the stats and value?

The one value for enterotype was calculated using chi-squared test as was stated before, now we further elaborate in Methods on lines 679-684:

"5.2.4. Enterotypes. Differences in the enterotype frequencies in PD vs NHC was tested using chi-squared test. The initial test was conducted on the distribution across three enterotypes in PD vs. NHC via `chisq.test` in R. To identify enterotypes that drive the difference, a sequential test was applied which uses chi-squared, tags the factor contributing the largest effect to chi-squared as the primary factor, removes the primary factor, normalizes the frequencies of remaining variables (enterotypes), repeats the process until chi-squared is no longer significant⁶⁴.

Per reviewer's comment, we now elaborate on and provide full stats for the enrichment of *Firmicutes* in Results lines 149-155:

"At enterotype level, 284 PD and 166 NHC were confidently classified as one of three enterotypes: *Prevotella*, *Firmicutes* and *Bacteroides* (Supplementary Fig. 2c). The overall enterotype distribution in PD was different from NHC ($P=1E-4$). Sequential testing identified *Firmicutes* enrichment as the primary driving force ($X^2=44.4$, $df=2$), and depletion of *Prevotella* as secondary as it remained significant ($X^2=3.7$, $df=1$) after removing *Firmicutes*. Effect sizes were $OR=2.4$ [1.5-3.9], $P=1E-4$, for *Firmicutes*, and a non-significant $OR=0.66$ [0.3-1.2] ($P=0.1$) for *Prevotella* after adjusting for *Firmicutes*' effect."

L139 -> Not entirely agree it is just species. One of the great applications of shotgun data is as well gene and functional annotation for further mechanism analysis.

Line 157: We changed “*A major goal of microbiome studies is to identify the disease-associated species*” to “Next, we attempted to identify the disease-associated species”

I suggest to revise the results subtitle and refer to the main outcome of the section. As it stands like “alignment with existing literature” or “replication” doesn’t really provide informative information of the section.

We have made the subtitles descriptive of the main outcomes, while keeping them at maximum 60 characters. The old titles (not highlighted) and new titles (highlighted) are:

The cohort = A large, newly enrolled, and uniformly assessed cohort

Metadata = Subject characteristics and metadata

Profiles of PD and NHC metagenomes = PD and NHC metagenomes differ from global scale to species-resolution

Network analysis reveals clusters of co-occurring and competing species = Network analyses reveal clusters of co-occurring & competing species (reduced to 60 characters)

Replication = Replication (no change)

Alignment with existing literature = Results align with existing literature and resolve inconsistencies

Complementarity of species-, genus- and cluster-level analyses = Species, genus, and cluster level analyses are complementary

Microbial gene-families and metabolic pathways = Altered abundances of microbial gene-families & metabolic pathways

Functional inference: Functional inference (no change)

Table R1. ANCOM-BC species MWAS (PD vs. NHC), with or without sequence count in the model, produce similar results. Both models include collection method as covariate.

Species	adjusted	for seq	not adjusted	for seq
	count		count	
	FDR	FC	FDR	FC
Actinomyces oris	2E-08	6.52	2E-08	6.49
Streptococcus mutans	3E-08	5.76	5E-08	5.70
Bifidobacterium dentium	8E-08	7.06	8E-08	7.06
Enterococcus avium	1E-06	2.60	1E-06	2.62
Blautia wexlerae	1E-05	0.20	1E-05	0.21
Ruminococcaceae bacterium D5	2E-05	4.18	2E-05	4.18
Clostridium leptum	3E-05	4.41	4E-05	4.38
Roseburia intestinalis	3E-05	0.13	4E-05	0.13
Streptococcus australis	2E-04	0.37	2E-04	0.38
Coprobacillus cateniformis	2E-04	2.37	2E-04	2.39
Ruminococcus lactaris	4E-04	0.17	4E-04	0.17
Anaerostipes hadrus	4E-04	0.19	4E-04	0.19
Fusicatenibacter saccharivorans	6E-04	0.19	6E-04	0.19
Turicibacter sanguinis	1E-03	2.57	1E-03	2.58
Lactobacillus salivarius	1E-03	2.31	1E-03	2.33
Eisenbergiella tayi	2E-03	3.61	2E-03	3.55
Ruthenibacterium lactatiformans	2E-03	1.89	2E-03	1.89
Scardovia wiggisiae	2E-03	1.65	2E-03	1.67
Eubacterium rectale	2E-03	0.24	2E-03	0.24
Faecalibacterium prausnitzii	3E-03	0.33	3E-03	0.33
Clostridium sp CAG 299	3E-03	0.29	3E-03	0.30
Ruminococcus callidus	3E-03	0.49	4E-03	0.50
Eubacterium sp CAG 38	3E-03	0.24	4E-03	0.24
Megasphaera sp DISK 18	4E-03	1.87	3E-03	1.90
Lactobacillus paragasseri	4E-03	1.82	4E-03	1.82
Lactobacillus fermentum	4E-03	2.70	4E-03	2.70
Actinomyces sp oral taxon 448	4E-03	1.63	4E-03	1.65
Actinomyces sp HPA0247	0.01	2.28	5E-03	2.28
Collinsella massiliensis	0.01	2.20	0.01	2.21
Porphyromonas asaccharolytica	0.01	1.64	0.01	1.64
Escherichia coli	0.01	4.03	0.01	4.02
Veillonella dispar	0.01	0.47	0.01	0.47
Ruminococcus bicirculans	0.01	0.23	0.01	0.23
Roseburia inulinivorans	0.01	0.28	0.01	0.28
Prevotella copri	0.01	0.27	0.01	0.27
Lactobacillus gasseri	0.01	1.68	0.01	1.69
Streptococcus mitis	0.01	0.48	0.01	0.48
Blautia hansenii	0.01	0.35	0.01	0.36
Streptococcus sp A12	0.01	0.59	0.02	0.60
Roseburia faecis	0.01	0.25	0.01	0.25
Actinomyces sp ICM47	0.01	0.40	0.01	0.40
Rothia mucilaginosa	0.01	0.41	0.01	0.41
Cloacibacillus evryensis	0.01	2.24	0.01	2.25
Methanobrevibacter smithii	0.01	3.26	0.01	3.29

Megasphaera sp MJR8396C	0.01	2.24	0.01	2.27
Hungatella hathewayi	0.01	2.90	0.02	2.87
Streptococcus anginosus group	0.01	2.06	0.02	2.05
Lactobacillus rhamnosus	0.02	2.15	0.02	2.15
Alistipes indistinctus	0.02	2.93	0.02	2.92
Streptococcus vestibularis	0.02	2.61	0.02	2.59
Dialister invisus	0.02	0.36	0.02	0.37
Acidaminococcus intestini	0.02	3.26	0.02	3.30
Klebsiella quasipneumoniae	0.02	2.27	0.02	2.28
Lachnoclostridium sp An131	0.02	1.64	0.02	1.64
Bifidobacterium gallinarum	0.02	1.45	0.02	1.46
Monoglobus pectinilyticus	0.02	0.32	0.02	0.32
Streptococcus infantis	0.02	0.64	0.03	0.64
Bifidobacterium pullorum	0.02	1.76	0.02	1.77
Enterococcus faecium	0.02	1.85	0.02	1.86
Eubacterium hallii	0.02	0.33	0.03	0.34
Butyricicoccus pullicaecorum	0.03	0.57	0.03	0.57
Klebsiella pneumoniae	0.03	2.73	0.03	2.72
Eubacterium ramulus	0.03	0.35	0.03	0.35
Eubacterium eligens	0.03	0.32	0.03	0.32
Bifidobacterium saeculare	0.03	1.38	0.03	1.40
Eubacterium limosum	0.04	1.67	0.04	1.69
Bifidobacterium bifidum	0.04	2.64	0.04	2.64
Bacteroides faecis CAG 32	0.04	1.92	0.04	1.94
Veillonella infantium	0.04	0.68	0.05	0.69
Actinomyces naeslundii	0.05	1.57	0.04	1.59
Enorma massiliensis	0.05	1.69	0.04	1.71
Eubacterium callanderi	0.05	1.73	0.04	1.76
Haemophilus sp HMSC71H05	0.05	0.70	0.05	0.70
Candidatus				
Methanomassiliicoccus				
intestinalis	0.05	1.58	0.04	1.60
Clostridium sp CAG 58	0.05	0.37	0.05	0.37
Bifidobacterium longum	0.05	2.74	0.05	2.76
Streptococcus lutetiensis	0.05	1.56	0.05	1.58
Lactobacillus reuteri	0.05	1.57	0.05	1.59
Eubacterium sp CAG 274	0.05	0.61	0.07	0.61
Ruminococcus torques	0.06	0.34	0.06	0.34
Haemophilus parainfluenzae	0.06	0.63	0.07	0.64
Clostridium sp CAG 253	0.07	0.65	0.07	0.65
Klebsiella variicola	0.07	2.11	0.07	2.10
Ruminococcus gnavus	0.07	0.41	0.07	0.41
Pseudoflavonifractor sp An184	0.07	1.60	0.07	1.59
Clostridium hylemonae	0.07	1.31	0.06	1.33
Firmicutes bacterium CAG 424	0.07	0.44	0.07	0.45
Clostridium innocuum	0.07	2.21	0.07	2.21
Blautia hydrogenotrophica	0.07	0.42	0.07	0.42
Agathobaculum				
butyriciproducens	0.07	0.39	0.07	0.39
Parabacteroides distasonis	0.08	2.19	0.08	2.18
Christensenella minuta	0.08	1.51	0.07	1.52

Harryflintia acetispora	0.09	1.69	0.09	1.69
Alistipes finegoldii	0.09	2.17	0.10	2.15
Clostridium bolteae CAG 59	0.09	0.51	0.10	0.52
Roseburia sp CAG 309	0.10	0.67	0.11	0.68
Clostridium sp CAG 167	0.10	0.58	0.11	0.59
Bacteroides coprocola	0.10	0.56	0.11	0.57
Blautia sp CAG 257	0.10	0.44	0.10	0.44
Bifidobacterium pseudocatenulatum	0.10	2.38	0.11	2.36
Eisenbergiella massiliensis	0.11	2.17	0.11	2.16
Bacteroides salyersiae	0.11	1.98	0.11	2.00
Paraprevotella clara	0.11	0.67	0.12	0.68
Oscillibacter sp CAG 241	0.12	2.09	0.12	2.09
Clostridium disporicum	0.12	1.47	0.11	1.49
Gemella sanguinis	0.12	0.66	0.13	0.66
Bifidobacterium breve	0.13	1.70	0.13	1.70
Veillonella atypica	0.13	0.62	0.14	0.63
Streptococcus thermophilus	0.15	0.50	0.15	0.50
Dorea longicatena	0.15	0.43	0.15	0.43
Anaerotignum lactatifermentans	0.15	0.51	0.15	0.51
Catabacter hongkongensis	0.16	1.40	0.16	1.40
Bifidobacterium adolescentis	0.16	2.20	0.16	2.21
Coprobacter fastidiosus	0.18	0.60	0.19	0.60
Faecalitalea cylindroides	0.18	1.52	0.18	1.52
Gordonibacter pamelaee	0.20	1.51	0.20	1.51
Barnesiella intestinihominis	0.20	2.07	0.21	2.06
Eubacterium sp CAG 251	0.20	0.65	0.23	0.66
Holdemanella biformis	0.20	0.65	0.22	0.66
Roseburia sp CAG 471	0.20	0.65	0.22	0.66
Paraprevotella xylaniphila	0.20	0.54	0.22	0.55
Lachnospira pectinoschiza	0.22	0.51	0.23	0.52
Lachnoclostridium sp An138	0.22	1.35	0.22	1.35
Enterococcus faecalis	0.23	1.33	0.22	1.34
Butyricimonas virosa	0.23	1.66	0.25	1.65
Coprobacter sp	0.24	0.70	0.25	0.71
Anaerofustis stercorihominis	0.25	1.30	0.23	1.31
Bacteroides faecis	0.26	1.70	0.25	1.73
Intestinimonas butyriciproducens	0.27	0.58	0.26	0.58
Actinomyces odontolyticus	0.27	0.65	0.27	0.65
Blautia sp An249	0.28	0.79	0.34	0.80
Pseudoflavonifractor capillosus	0.31	0.80	0.31	0.80
Blautia producta	0.32	1.67	0.31	1.69
Bacteroides galacturonicus	0.33	0.77	0.35	0.78
Bacteroides cellulosilyticus	0.34	1.79	0.34	1.77
Bacteroides caccae	0.34	1.84	0.34	1.84
Firmicutes bacterium CAG 94	0.34	1.65	0.34	1.64
Actinomyces johnsonii	0.35	1.19	0.33	1.20
Parabacteroides goldsteinii	0.35	1.57	0.34	1.58
Bacteroides vulgatus	0.36	0.71	0.34	0.71
Propionibacterium freudenreichii	0.36	0.83	0.41	0.84
Parabacteroides merdae	0.36	1.69	0.37	1.68

Ruminococcus bromii	0.37	0.55	0.37	0.55
Alloscardovia omnicolens	0.37	1.23	0.34	1.25
Lawsonibacter asaccharolyticus	0.38	0.68	0.37	0.68
Parabacteroides johnsonii	0.38	0.73	0.41	0.74
Blautia coccoides	0.39	1.50	0.38	1.51
Turicimonas muris	0.39	0.68	0.38	0.68
Bacteroides xylanisolvens	0.39	1.64	0.39	1.64
Streptococcus oralis	0.39	0.77	0.40	0.77
Cloacibacillus porcorum	0.40	1.22	0.38	1.23
Akkermansia muciniphila	0.40	1.66	0.41	1.64
Firmicutes bacterium CAG 145	0.41	1.48	0.41	1.48
Clostridium bolteae	0.41	0.67	0.41	0.67
Firmicutes bacterium CAG 95	0.42	0.82	0.46	0.83
Roseburia hominis	0.42	0.64	0.43	0.64
Lactobacillus animalis	0.45	1.24	0.43	1.24
Bacteroides plebeius	0.45	0.68	0.45	0.68
Streptococcus sanguinis	0.45	0.79	0.46	0.79
Proteobacteria bacterium CAG 139	0.45	0.66	0.45	0.65
Collinsella stercoris	0.45	1.41	0.44	1.43
Eubacterium sp CAG 180	0.47	0.69	0.48	0.69
Bacteroides nordii	0.47	0.81	0.50	0.82
Clostridium symbiosum	0.47	1.47	0.48	1.46
Flavonifractor plautii	0.51	0.78	0.50	0.78
Eggerthella lenta	0.51	1.40	0.50	1.40
Collinsella aerofaciens	0.51	1.42	0.51	1.42
Lactobacillus plantarum	0.51	1.18	0.48	1.19
Oscillibacter sp 57 20	0.52	0.65	0.53	0.66
Actinomyces sp HMSC035G02	0.52	0.80	0.52	0.80
Bacteroides thetaiotaomicron	0.52	1.43	0.53	1.42
Bilophila wadsworthia	0.52	1.39	0.53	1.39
Clostridium saccharolyticum	0.52	1.30	0.50	1.32
Phascolarctobacterium succinatutens	0.53	1.24	0.50	1.25
Clostridium methylpentosum	0.53	1.20	0.50	1.22
Anaerostipes caccae	0.54	0.77	0.54	0.77
Candidatus Stoquefichus sp KLE1796	0.59	0.80	0.61	0.81
Bacteroides intestinalis	0.59	1.35	0.58	1.36
Bacteroides eggerthii	0.60	0.73	0.60	0.73
Bacteroides massiliensis	0.61	1.37	0.60	1.37
Clostridium scindens	0.61	1.31	0.60	1.32
Clostridiales bacterium 1 7 47FAA	0.61	0.81	0.62	0.81
Lactobacillus rogosae	0.62	0.88	0.63	0.88
Anaerotruncus colihominis	0.62	1.31	0.63	1.29
Anaeromassilibacillus sp An250	0.62	1.28	0.63	1.27
Clostridium aldenense	0.62	0.80	0.63	0.81
Clostridium sp CAG 242	0.62	0.80	0.63	0.80
Bacteroides stercoris	0.62	1.35	0.66	1.32
Bifidobacterium animalis	0.62	0.83	0.66	0.84

Clostridium lavalense	0.62	0.78	0.63	0.77
Odoribacter splanchnicus	0.62	1.35	0.63	1.35
Eubacterium siraeum	0.63	0.73	0.63	0.73
Actinomyces sp oral taxon 181	0.65	1.11	0.64	1.11
Lactobacillus vaginalis	0.66	1.09	0.63	1.11
Erysipelatoclostridium ramosum	0.69	0.80	0.70	0.81
Romboutsia ilealis	0.70	1.07	0.64	1.08
Holdemania filiformis	0.71	1.21	0.71	1.20
Dialister sp CAG 357	0.71	1.15	0.70	1.15
Blautia sp N6H1 15	0.71	0.87	0.73	0.88
Bacteroides uniformis	0.71	0.88	0.70	0.88
Bacteroides fragilis	0.71	0.77	0.70	0.77
Phascolarctobacterium faecium	0.71	1.26	0.70	1.28
Butyricimonas synergistica	0.72	1.19	0.73	1.18
Clostridium asparagiforme	0.72	1.21	0.72	1.21
Murimonas intestini	0.72	1.10	0.70	1.12
Absiella dolichum	0.73	0.87	0.75	0.88
Dorea sp CAG 317	0.73	0.80	0.73	0.80
Desulfovibrio piger	0.73	1.11	0.71	1.12
Bacteroides finegoldii	0.74	1.22	0.74	1.21
Blautia obeum	0.74	0.81	0.74	0.81
Firmicutes bacterium CAG 110	0.74	1.16	0.72	1.17
Lactococcus lactis	0.74	1.17	0.76	1.16
Collinsella intestinalis	0.75	1.20	0.73	1.21
Odoribacter laneus	0.75	1.11	0.73	1.13
Mogibacterium diversum	0.77	1.07	0.74	1.07
Eubacterium dolichum CAG 375	0.77	0.89	0.80	0.90
Coprococcus comes	0.77	1.23	0.76	1.24
Coprococcus eutactus	0.77	0.89	0.78	0.89
UNKNOWN	0.77	0.97	0.79	0.97
Prevotella corporis	0.77	1.07	0.76	1.07
Clostridium spiroforme	0.78	1.10	0.78	1.10
Dielma fastidiosa	0.82	1.10	0.82	1.10
Asaccharobacter celatus	0.83	0.89	0.83	0.89
Bacteroides dorei	0.83	1.18	0.83	1.17
Alistipes onderdonkii	0.84	0.93	0.85	0.94
Streptococcus parasanguinis	0.84	0.90	0.83	0.90
Ruminococcaceae bacterium D16	0.84	1.10	0.83	1.11
Coprococcus catus	0.84	0.88	0.85	0.89
Catenibacterium mitsuokai	0.84	1.07	0.82	1.08
Rothia dentocariosa	0.86	1.07	0.85	1.07
Adlercreutzia equolifaciens	0.87	0.91	0.85	0.91
Allisonella histaminiformans	0.87	0.96	0.89	0.97
Enterorhabdus caecimuris	0.87	0.94	0.85	0.94
Gemmiger formicilis	0.87	0.90	0.85	0.90
Oxalobacter formigenes	0.87	1.06	0.85	1.07
Actinomyces graevenitzii	0.87	0.94	0.87	0.94
Eubacterium ventriosum	0.87	1.07	0.85	1.09
Clostridium citroniae	0.88	1.08	0.88	1.08
Slackia isoflavoniconvertens	0.89	1.06	0.85	1.07

Flavonifractor sp An100	0.89	1.03	0.85	1.04
Alistipes putredinis	0.89	1.09	0.90	1.08
Alistipes shahii	0.89	1.08	0.90	1.08
Lactonifractor longoviformis	0.89	1.04	0.88	1.05
Sutterella parvirubra	0.89	1.03	0.85	1.05
Alistipes inops	0.89	1.04	0.87	1.06
Intestinibacter bartlettii	0.90	0.94	0.90	0.94
Veillonella parvula	0.91	1.04	0.91	1.04
Firmicutes bacterium CAG 83	0.92	0.94	0.93	0.95
Streptococcus gordonii	0.92	0.96	0.91	0.96
Dorea formicigenerans	0.93	0.95	0.94	0.96
Parasutterella excrementihominis	0.93	1.05	0.95	1.03
Clostridium clostridioforme	0.94	0.96	0.95	0.96
Sellimonas intestinalis	0.94	1.04	0.95	1.04
Alistipes timonensis	0.94	1.01	0.92	1.02
Tyzzereella nexilis	0.99	1.00	0.99	1.00
Bacteroides ovatus	0.99	0.99	0.97	0.98
Desulfovibrio fairfieldensis	0.99	1.00	0.95	1.01
Streptococcus salivarius	1.00	1.00	0.99	0.99

We hope we understood the comments thoroughly and responded satisfactorily.

Thank you for your time
Haydeh Payami

REVIEWERS' COMMENTS

Reviewer #1 (Remarks to the Author):

The authors have addressed my comments, except of that regarding the association between sporulating bacteria-EC serotonin production and gut motility. While endogenous 5-HT is released in response to a number of stimuli and plays an important role in paracrine and endocrine systems, it acts only as a modulator, not as an initiator, of neurogenic motor patterns and gut transit (<https://doi.org/10.1016/j.phrs.2018.06.017>). Accordingly, I suggest that the authors should dampen their hypothesis even if it based on data and association because the assumption is not precise in the first place.

Reviewer #2 (Remarks to the Author):

The paper was consistently improved. The comments were addressed appropriately.

Here are some minor issues with Fig. 2:

Fig 2a: It seems the axis labels are a mixture of linear scale and log scale. If it is in linear scale, it should range from 1 to 0 (the starting point should be 1); if it is in negative log scale, it should range from 0 to positive infinity.

Fig 2b and c: legends say that 2b is for $FDR \leq 0.05$, and 2c is for $FDR \leq 0.1$, which means we should have more significant taxa shown in 2c, right? However, clearly, there are more significant taxa in 2b, perhaps there is a potential label flip?

Metagenomics of Parkinson's disease implicates the gut microbiome in multiple disease mechanisms (NCOMMS-22-22904A)

Point-by-Point
October 27, 2022

We very much appreciate the time you have spent on this manuscript and your helpful reviews. We have pasted the reviews below (in black font), with our responses (in blue). Changes made to the manuscript are indicated by line number and are highlighted grey.

REVIEWERS' COMMENTS

Reviewer #1 (Remarks to the Author):

The authors have addressed my comments, except of that regarding the association between sporulating bacteria-EC serotonin production and gut motility. While endogenous 5-HT is released in response to a number of stimuli and plays an important role in paracrine and endocrine systems, it acts only as a modulator, not as an initiator, of neurogenic motor patterns and gut transit (<https://doi.org/10.1016/j.phrs.2018.06.017>). Accordingly, I suggest that the authors should dampen their hypothesis even if it based on data and association because the assumption is not precise in the first place.

Thank you for clarifying that the issue is initiation vs. modulation. I agree, we were not being precise about initiation vs modulation. We have modified the text and added the reference you provided. On line 451-452 "It was shown in mice that spore forming bacteria induce production of serotonin in host colonic cells⁴⁹" we deleted "which stimulates myenteric neurons and gut motility". Instead, we start the next paragraph with (Line 460-461): "Serotonin affects gut motility through modulation of intestinal contractility⁵⁰. Given that spore forming bacteria modulate serotonin production and enhance gut motility^{49,50}..." ref 50 is (<https://doi.org/10.1016/j.phrs.2018.06.017>)," On line 796, in "prior studies have shown sporulation species modulate gut motility⁴⁹" we replaced "are necessary for" with "modulate".

Reviewer #2 (Remarks to the Author):

The paper was consistently improved. The comments were addressed appropriately.

Here are some minor issues with Fig. 2:

Fig 2a: It seems the axis labels are a mixture of linear scale and log scale. If it is in linear scale, it should range from 1 to 0 (the starting point should be 1); if it is in negative log scale, it should range from 0 to positive infinity.

Axes are negative log. We had the actual FDR that corresponds to the -log for easier interpretation, as we have done in other figures, but we neglected to point it out here. We added the actual value of $-\log(\text{FDR})$ to the graph followed by the untransformed FDR in parentheses. In legend, we added "Corresponding untransformed FDR values are provided in parentheses for easier interpretation." Line 1106.

Fig 2b and c: legends say that 2b is for $\text{FDR} \leq 0.05$, and 2c is for $\text{FDR} \leq 0.1$, which means we should have more significant taxa shown in 2c, right? However, clearly, there are more significant taxa in 2b, perhaps there is a potential label flip?

Yes, they were flipped in the legend. We corrected it: Line 1114.

Thank you sincerely
Haydeh Payami